# How horizontal transport and turbulent mixing impacts aerosol particle and precursor concentrations at a background site in the UAE

Jutta Kesti[1], Ewan J. O'Connor[1], Anne Hirsikko[1], John Backman[1], Maria Filioglou[2], Anu-Maija Sundström[1], Juha Tonttila[2], Heikki Lihavainen[1,3], Hannele Korhonen[1], and Eija Asmi[1]

[1]Finnish Meteorological Institute, Helsinki, Finland
[2]Finnish Meteorological Institute, Kuopio, Finland
[3]Svalbard Integrated Arctic Earth Observing System, Longyearbyen, Norway

**Correspondence:** Jutta Kesti (jutta.kesti@fmi.fi)

**Abstract.** Aerosol particle optical, physical and chemical properties have been previously studied in the United Arab Emirates (UAE), but there is still a gap in the knowledge of particle sources, and in the horizontal and vertical transport of aerosol particles and their precursors in the area. To investigate how aerosol particle and $SO_2$ concentrations at the surface responded to changes in horizontal and vertical transport, we used data from a one-year measurement campaign at a background site where local sources of $SO_2$ were expected to be minimal. The measurement campaign provided a combination of in-situ measurements at the surface, and the boundary layer evolution from vertical and horizontal wind profiles measured by a Doppler lidar. The diurnal structure of the boundary layer in the UAE was very similar from day to day, with deep well-mixed boundary layer during the day transitioning to a shallow nocturnal layer, with the maximum boundary layer height usually being reached around 1400 local time. Both $SO_2$ and nucleation mode aerosol particle concentrations were elevated for surface winds coming from the east or western sectors. We attribute this to oil refineries located on the eastern and western coasts of the UAE. The concentrations of larger cloud condensation nuclei (CCN) sized particles and their activation fraction did not show any clear dependence on wind direction, but the CCN number concentration showed some dependence on wind speed, with higher concentrations coinciding with the weakest surface winds. Peaks in $SO_2$ concentrations were also observed despite low surface wind speeds and wind directions unfavourable for transport. However, winds aloft were much stronger, with wind speeds of $10\ \mathrm{m\,s^{-1}}$ at 1 km common at night, and with wind directions favourable for transport, and surface-measured concentrations increased rapidly once these particular layers started to be entrained into the growing boundary layer, even if the surface wind direction was from a clean sector. These conditions also displayed higher nucleation mode aerosol particle concentrations, i.e. new particle formation events occurring due to the increase in the gaseous precursor.

## 1 Introduction

Aerosol particles have various climate impacts and need to be carefully studied to provide more accurate future climate predictions. Aerosol particles originate from both natural and anthropogenic sources. The smallest aerosol particles can form directly

in the atmosphere through nucleation and growth of gaseous precursors (Kulmala et al., 2004) in a process named as new particle formation (NPF) event. One of the most important NPF precursors in the atmosphere is sulphuric acid ($H_2SO_4$), forming through photochemical reactions from $SO_2$ (Weber et al., 1997; Kulmala et al., 2000; Sipilä et al., 2010). A fraction of the ultrafine aerosol particles formed in the NPF events can further grow and act as cloud condensation nuclei (CCN) (Laaksonen et al., 2005; Kuang et al., 2009; Merikanto et al., 2009; Bzdek and Johnston, 2010). The CCN are important because of their ability to form cloud droplets, and their concentrations have been shown to modify cloud properties such as the lifetime of clouds and hence having impact on cloudiness and rainfall patterns (Albrecht, 1989; Jiang et al., 2006). To understand the complex chain of aerosol particle processes which are critical for the climate, it is important to study the combination of $SO_2$ concentration, newly formed aerosol particles and CCN concentration in the atmosphere.

Horizontal and vertical air transport constantly modify the aerosol particle population we measure at fixed surface location and the transport phenomena connected with the local meteorology are important factors to consider to interpret the aerosol particle observations. Derimian et al. (2017) observed in the Neger Desert of Israel a shift towards larger aerosol particle sizes in the aerosol particle size distribution, which was associated with the sea breeze arrival. Krishna Moorthy et al. (2003) also observed a shift towards larger sizes in the aerosol particle mass size distribution when the air mass changed from continental to marine at a remote coastal station near the south-west tip of the Indian Peninsula. Kumar et al. (2016) measured the highest mass concentrations of elemental carbon, organic carbon and water-soluble organic carbon at a high-altitude site (Mt Abu) in western India during winter months and attributed it "to the synoptic-scale long-range transport from the emission source regions in the Indo-Gangetic Plain". Zhang et al. (2009) found 3 different types of aerosol particle vertical distributions which were associated with different meteorological and weather conditions in Beijing, China. Wagner et al. (2009) observed increased aerosol particle concentrations compared to the surface in an elevated plume of Saharan dust transported to Portugal and hence underlined the importance of multi-platform measurements. The findings from previous studies underline the importance of studying aerosol particles in context with the potential for transport, suggesting that the profile of the horizontal wind should be measured, as should the profile of turbulent mixing or the identification of the well-mixed boundary layer.

The primary aerosol particles in the Arabian Peninsula originate from natural sources (desert dust, sea salt) and anthropogenic sources (road traffic, petroleum industry and construction activities, Khodeir et al., 2012; Semeniuk et al., 2015; Lihavainen et al., 2016; Rushdi et al., 2017; Wehbe et al., 2021). In addition, the smaller secondary aerosol particles are also play a key role in the region according to previous studies. Exceptionally frequent and strong NPF events were observed in Hada Al Sham, Saudi Arabia, by Hakala et al. (2019). They attributed the NPF events to the transportation of precursor vapors, especially sulphuric acid, from coastal cities and industrial areas by the developed sea breeze in daytime. Frequent NPF events were also found in the UAE by Kesti et al. (2022).

There are only a few previous studies describing aerosol particle optical properties in the UAE. Filioglou et al. (2020) reported the optical and geometrical properties of aerosol particles in the UAE region suggesting "that the pure dust properties over the Middle East and western Asia, including the observation site, are comparable to those of African mineral dust regarding the linear particle depolarization ratios, but not for the lidar ratios". Long spatiotemporal trends of AOD in the UAE region were studied by Abuelgasim et al. (2021) and they found a significant increase in AOD during summer due to high wind

speeds connected to primary aerosol particle sources. Beegum et al. (2016) investigated optical and radiative properties of aerosol particles over Abu Dhabi and found that AOD spectra varied "significantly throughout the year with higher aerosol loading and flatter spectra during spring/summer seasons and comparatively steeper spectra and lower values of AOD during autumn/winter seasons". The physical and chemical properties of aerosol particles and their response to mixing in the boundary layer were studied by Kesti et al. (2022) at a background site in the UAE. They concluded that the vertical mixing of aerosol particles and their precursors potentially generate horizontal layers that could either favor or hinder secondary aerosol particle formation. Yet, aerosol particle and CCN regional sources and the particle transport mechanisms and dynamics have not been extensively studied in the UAE region.

Water shortage is a serious threat in the UAE and the larger Arabian Peninsula region (Wehbe et al., 2018; Wehbe and Temimi, 2021). Cloud seeding as one of the precipitation enhancement techniques has been under investigation as part of UAE's strategy to solve water shortage issues (Al Hosari et al., 2021; Wehbe, 2022). The Optimization of Aerosol Seeding In rain enhancement Strategies (OASIS) project aimed to characterize the properties and efficiency of aerosol particles to act as CCN in the UAE region. In this study we focus on how air mass transport and boundary layer mixing affect the aerosol particle and CCN concentrations measured at the surface. Typical methods for studying the vertical profile of aerosol particle properties are balloon sounding campaigns or aircraft measurements, but these methods are expensive and only cover short time periods. Here, we use continuous measurements of the lowest 2 km of the atmosphere with a ground-based Doppler lidar. This will help us to understand the surface/aerosol exchange in the area which is not that well known. In section 2 we give a brief description of the measurement site and methods used. In section 3 we analyse how horizontal and vertical transport affect aerosol particle properties and $SO_2$ concentration at the surface.

## 2   Instrumentation and methods

### 2.1   Site description

The measurement campaign was conducted on a palm tree farm ($25°14'7.8''$ N, $55°58'39.97''$ E, 165 metres above sea level, Filioglou et al., 2020) representing rural background with mainly sand desert and agricultural activities in the surroundings (Wehbe et al., 2017). The Arabian Gulf is about 70 km west of the measurement site and the Gulf of Oman around 40 km to the east. The western coastline hosts the city of Dubai and at least two major oil refineries. Fujairah City and one oil refinery are located on the eastern coast (Fig. 1).

### 2.2   Data sets

The measurement campaign was performed between February 2018 and February 2019. The campaign, instrumentation deployed and data sets are described in more detail in Kesti et al. (2022). In this study we used the near-surface winds and the vertical profile of wind and mixing to investigate the measured concentrations and variability of aerosol particle nucleation

mode concentration, $SO_2$ concentration, CCN concentration and activation fraction. Here, we briefly summarise the main measurement methodology.

The surface meteorological parameters were measured with an automatic weather station (Vaisala WXT 520) with a 5-minute time resolution. The sensor was 7 meters above ground level.

The vertical profile of horizontal wind speed and direction were measured with a HALO Photonics Streamline Doppler lidar with a time resolution of 15 min. The profile of turbulent mixing was retrieved using the boundary layer classification described by Manninen et al. (2018), which combines the profiles of attenuated backscatter coefficient, vertical velocity skewness, dissipation rate, horizontal wind, and vector wind shear measured by the HALO Doppler lidar. The boundary layer classification scheme identifies the regions where mixing is connected to the surface, from which the mixing layer height is retrieved. The temporal resolution of the classification is 3 min. Detailed instrument specifications, instrument scan schedule, and data processing methods are discussed in Kesti et al. (2022).

The aerosol particle size distribution from size 7 nm to 800 nm was measured with a Differential Mobility Particle Sizer (DMPS). The size range was divided to 30 discrete size bins, and the size spectra was measured over 6 min and 25 s. The data was inverted with the method described by Aalto et al. (2001) and Wiedensohler et al. (2012). In this study we focused on the nucleation mode aerosol particle concentration from 10 nm to 25 nm.

The $SO_2$ concentrations were measured with a Thermo Scientific 43i-TLE. The time resolution for the measurement was 30 s.

The CCN concentration was measured with the Cloud Condensation Nuclei counter (CCNc, Droplet Measurement Technology, Roberts and Nenes, 2005). The CCNc was operated at five different supersaturations (0.1, 0.2, 0.3, 0.6, 1.0), and in this study we focused on the measurements with the supersaturation 1.0. The supersaturation 1.0 was chosen to investigate all the possible aerosol particles that would activate to CCN, both the very hygroscopic and less hygroscopic. The time resolution of the measurements was 60 min. To provide size-resolved CCN measurements, a differential mobility analyser (DMA), was added to enable the CCNc to measure activated fraction of aerosol particles as a function of size (size range 10 – 250 nm). The CCN number concentration was compared to the number concentration measured with a condensation particle counter (CPC) to determine the CCN-activated fraction of aerosol particles at a certain size. A detailed explanation of the calibration procedures performed for this instrument is given in Kesti et al. (2022).

Air mass back-trajectories were computed using the Hybrid Single Particle Lagrangian Integrated Trajectory (HySPLIT) model (Stein et al., 2015) with GDAS1 meteorological information. A set of 48-hour backward trajectories were computed for every hourly-derived HALO Doppler lidar mixing layer height, each set comprising one back-trajectory starting every 200 m in altitude from the surface to the top of the mixing layer at that hour. Then, these trajectories were collocated in time with $SO_2$ observations and further separated into Low ($SO_2$ <= 0.1019 ppb) and High ($SO_2$ >= 1.1740 ppb) concentrations utilising the 10-90th percentile ratio from the $SO_2$ observations. The trajectories were further binned onto a horizontal 0.5 x 0.5° grid keeping the counts passing through each grid cell to further derive the relative difference (RD) expressed as RD = (H-L)/(H+L) for every grid cell that the trajectories have passed through.

## 3 Results and discussion

Previous studies (Filioglou et al., 2020; Kesti et al., 2022) have suggested clear diurnal and seasonal temporal patterns in atmospheric transport at the site. The seasonal variability in atmospheric conditions is quite weak with the main difference between summer (March-August) and winter (September-February) being slightly higher wind speeds during summer ($2.3 \pm 1.6 \mathrm{~m~s^{-1}}$ in summer and $2.1 \pm 1.5 \mathrm{~m~s^{-1}}$ in winter). The diurnal variability in surface wind direction is very distinctive and predictable, with winds being easterly before sunrise, turning to a western and north-western direction during the day, and then back to being easterly at night. This reliable pattern allows us to investigate how changes in aerosol particle and $SO_2$ concentrations are linked to local and synoptic scale transport.

### 3.1 Observed meteorological patterns

The diurnal and seasonal behaviour of the surface meteorological parameters measured at the site were analysed in Kesti et al. (2022). The observed surface temperatures varied from 10 to 48 °C and the highest temperatures were at noon during summer. The surface relative humidity varied from 6 to 90 % with the minimum during summer when the temperature reached its maximum. Ambient pressure was stable during the campaign. There were only eight rain events that were observed at the surface. The wind speeds were generally low at the surface as evidenced by the mean hourly wind speed of $2.2 \mathrm{~m~s^{-1}}$. The maximum wind speed of $17.5 \mathrm{~m~s^{-1}}$ was measured on 13 May 2018 and the highest wind speeds were mostly observed in early summer. In the aforementioned study at this site, only the dominant wind conditions on an hourly or monthly basis were presented. Here, using wind roses, the seasonal and diurnal changes in the full wind distribution were investigated with respect to local time (UTC + 4). Our results indicate a clear shift in the surface winds over the course of the year (Fig. 2); during autumn and winter months the wind from the east was more frequent whereas in late spring and early summer the wind from the west was more frequent. Prevailing westerly winds were observed during spring and early summer. Winds from the east were generally associated with low wind speeds.

A clear diurnal cycle in the near-surface wind can been seen in Fig. 3, where easterly winds at night change to westerly or north-westerly during the day, and back to easterly by late evening. The winds at night are usually very light with stronger winds present during the day, whether from the west or east. Wind roses for every hour separately can be seen in Fig. A1. The seasonal variation in wind was also investigated.

Following Filioglou et al. (2020), the seasons were divided into two: summer from March to August; winter from September to February. During the summer months there was more airflow from the west compared to the winter months, when the predominant wind direction was the east (Fig. A2). The observed diurnal variation in the wind direction at the site reflects the wind patterns that are so typical for the Arabian Gulf. Recent reanalysis of meteorological data over the past 40 years shows that the surface winds over the Arabian Gulf are predominantly north-westerly throughout the year (Dasari et al., 2022; Patlakas et al., 2019). In addition to this, there is a land-sea breeze that has been described as perennial and can reach 100's of km inland, except for the northern parts of the gulf (Zhu and Atkinson, 2004; Eager et al., 2008). According to Eager et al. (2008), the sea breeze is usually first observed at the coast at midday local time and then moves inland, being observed 15-20

km inland by about 1500 local time. A land-breeze occurring at night and flowing in the opposite direction to the daytime sea breeze is also evident in the climatological reanalysis. The HALO Doppler lidar wind profiles show that there is significant transport aloft (from the direction of the gulf) throughout the early morning and during the day, which corresponds to the larger mesoscale wind direction (north-westerly flow) for the region which seems to be persistent all year round (Eager et al., 2008; Zhu and Atkinson, 2004). For a global picture on the links between the the Arabian peninsula's climate and global circulation patterns, see e.g. Attada et al. (2018) and references therein.

## 3.2 Daily evolution of boundary layer and vertical wind patterns

Kesti et al. (2022) investigated the effects of turbulent and non-turbulent boundary layer conditions and also effects of boundary layer height on aerosol particle properties and $SO_2$ concentrations. They used three case studies: 1. Deep boundary layer with horizontal transport aloft, 2. Shallow boundary layer and 3. Deep boundary layer, stagnant residual layer aloft. In this study, we investigate the boundary layer evolution and horizontal transport more broadly.

We defined the hours when boundary layer height reached its maximum during each day during the campaign. The evolution of the boundary layer was analysed between sun rise and sun set hours, 5 am – 8 pm. In our analysis we used the boundary layer classification method developed by Manninen et al. (2018) to diagnose when surface-connected mixing was present. The number of days when surface connected mixing was occurring at the site was around 90 % of all campaign days. Figure 4 shows that the maximum boundary layer height is most often reached after noon around 1 pm - 3 pm.

The length of the day only varies by 3 hours in the UAE region and the diurnal structure of the boundary layer is very similar from day to day (Fig. 5). The median boundary layer height during the campaign was around $265 \pm 311$ m at night and around $1067 \pm 894$ m during the day.

Figure 5 shows that, with a shallow nocturnal boundary layer and light winds, the horizontal transport within the boundary layer (and hence the surface) at night is slow. However, there are stronger horizontal winds above at night, providing significant transport. Moreover, the wind direction can be different from the surface wind direction. Once the morning boundary layer grows to the altitudes where more significant $SO_2$ transportation occurs, we observe peaks in $SO_2$ concentration at the surface once these elevated layers are mixed down to the surface. Indications of this mixing of pollutants from above down to the surface was already reported by Kesti et al. (2022) in a case study of a deep boundary layer with a stagnant residual layer aloft.

## 3.3 Effect of horizontal and vertical air mass transport on $SO_2$ concentration and aerosol particle properties

In the next section, we investigate how the observed vertical and horizontal air mass transport, and mixing, impact aerosol particle processes, focusing on the formation of nucleation mode aerosol particles from precursors, here mainly $SO_2$ (Marti et al., 1997; Birmili and Wiedensohler, 2000).

As discussed in Section 3.1, the diurnal variation in wind speed and direction was very distinctive in the study area and this is why we divided the data to daytime (5 am – 8 pm local time) and nighttime (8 pm – 5 am local time). Figure 6 shows the difference between daytime and nighttime wind speeds. During daytime, the wind speed distribution is wider and the speeds are more evenly distributed whereas during nighttime the wind speed is generally quite low.

### 3.3.1 SO$_2$ and nucleation mode aerosol particles

We measured higher SO$_2$ concentrations when the wind was coming from specific directions during daytime (Fig. 7a). For example, when considering mean concentrations from wind directions 20–40°, 80–100°, 170–190° and 270–290° in daytime, the measured mean concentrations were $0.34 \pm 0.32$ ppb, $0.24 \pm 0.32$ ppb, $0.37 \pm 0.66$ ppb and $1.13 \pm 1.88$ ppb respectively. There are at least four major oil refineries known to us around the measurement site: one in the east, two to the southwest and one to the northwest (Fig. 1). The locations of the refineries correlate with the observed elevated SO$_2$ concentrations when the wind was coming from eastern or western directions. Figure 11(a) shows average SO$_2$ column density number in Dobson units retrieved from the TROPOMI satellite measurements from May 2018 until February 2019. Elevated SO$_2$ values can be seen in Dubai compared to the surroundings which is in agreement with our surface measurements. Other refinery locations are not that clearly recognisable. It should be noted that the satellite measurements and surface measurements are not directly comparable, since the satellite measurement is integrated through the entire atmospheric column. When the airflow was coming from the west in daytime, we measured higher SO$_2$ concentrations compared to eastern direction. When considering wind direction sectors 45–135° for the east and 225–315° for the west, the measured mean concentrations were $0.26 \pm 0.36$ ppb and $1.00 \pm 1.63$ ppb respectively. One explanation is the higher number of oil refineries on the western coast but also the mountains on the eastern side which could deflect the SO$_2$ pollution from reaching the site. However, there is a clear increase in SO$_2$ concentrations when the wind is from the east. The airflow from the east is more predominant in nighttime and hence the measured elevated SO$_2$ concentration in nighttime is consistent with the wind direction where the source for SO$_2$ exists (Fig. 7b). In nighttime the measured mean SO$_2$ concentrations from wind directions 20–40°, 80–100°, 170–190° and 270–290° were $0.47 \pm 1.05$ ppb, $0.23 \pm 0.27$ ppb, $0.38 \pm 0.71$ ppb and $0.46 \pm 0.71$ ppb respectively. In daytime there is no clear dependence of SO$_2$ concentration on wind speed (Fig. 7c). In nighttime the concentrations are lower at higher wind speeds (Fig. 7d) but as Fig. 6 shows, there are not many high wind speed cases during night (average concentrations in wind speeds 0–2 m s$^{-1}$ and higher than 2 m s$^{-1}$ were $0.31 \pm 0.46$ ppb and $0.30 \pm 0.53$ ppb respectively having almost four times more data points in the first wind speed bin).

Figure 8a shows an increase in nucleation mode aerosol particle number concentration during daytime compared to nighttime (Fig. 8b). The mean nucleation mode aerosol particle concentration during daytime was $1008 \pm 1802$ cm$^{-3}$ and during nighttime $330 \pm 338$ cm$^{-3}$. The clearly higher nucleation mode aerosol particle concentrations in daytime are explained by solar radiation which through photochemistry causes new particle formation. The increased concentrations are focused on eastern and western wind directions, with higher concentrations in the western airflow. In daytime the mean nucleation mode aerosol particle concentrations in wind direction sectors 20–40°, 80–100°, 170–190° and 270–290° were $719 \pm 859$ cm$^{-3}$, $517 \pm 851$ cm$^{-3}$, $519 \pm 1192$ cm$^{-3}$ and $2096 \pm 2469$ cm$^{-3}$ respectively. This result is consistent with the SO$_2$ concentration (Fig. 7a), which is supporting the fact that one of the most important new particle formation precursors in the atmosphere is sulfuric acid (H$_2$SO$_4$), which forms from sulfur dioxide (Weber et al., 1997; Kulmala et al., 2000; Sipilä et al., 2010). In nucleation mode aerosol particles the effect of wind speed is following the same pattern than in SO$_2$ concentration: in daytime

there is no clear dependence of nucleation mode aerosol particle concentration on wind speed (Fig. 8c), and during nighttime (Fig. 8d) the distribution is following the shape of the nighttime wind speed distribution in Fig. 6.

### 3.3.2    Cloud Condensation Nuclei (CCN)

The effects of wind direction and time of the day were seen in nucleation mode aerosol particle and $SO_2$ concentrations. More specifically, winds from the west are associated with both increased $SO_2$ concentrations and nucleation mode particles (Figs.
7 and 8). Westerly winds (at ground level) is the typical wind direction during the day, and the nighttime wind direction is typically from the east; see Fig. 3. When plotting the activated fraction of particles as a function of wind direction, the pattern is not as clear as for $SO_2$ and nucleation mode aerosols (Fig. 9). The dotted line in Fig. 9) shows the critical diameter ($D_{50}$) at which 50% of the particles have activated. Even at a supersaturation of 1.0, $D_{50}$ is well above the nucleation mode size range, and was on average 37.9 nm for all wind directions. The mean $D_{50}$ for 0.6, 0.3 and 0.2 % were 47.6, 67.5 and 99.1
nm, respectively. The $D_{50}$ values were calculated from fitting a sigmoid curve to the CCN/CN data for each DMA scan (from 10 to 250 nm) and using that fit to determine where the curve passed 50% activation of CCN compared CN for the different supersaturations. The lowest supersaturation 0.1 was omitted since that data was found to be unstable.

    Activation fraction (CCN/CN) was slightly higher (around 0.6) when the wind came from the east whereas the activation fraction was a bit lower when the wind came from the west (around 0.4, Fig. 10a). This could be due to different sources of
aerosol particles from the different areas. More aerosol particles are coming from the west but they do not activate that easily (comparing Fig. 10a with Fig. 8a). There was no clear difference between daytime and nighttime in activation fraction, but the absence of wind from western direction and hence the other source of CCN can be seen in Fig. 10b. Wind speed did not have a clear effect on activation fraction at the site (Fig. 10c and d).

    The figure shows that the $D_{50}$ range between around 40 nm – 100 nm at the site. Table A1 shows more statistics for the
activation fraction and also calculated $\kappa$ values at different supersaturations.

### 3.3.3    Effect of boundary layer evolution on $SO_2$

Horizontal transport of $SO_2$ at the surface is not sufficient for explaining the variation in the concentrations measured at the surface during the campaign; in conditions with low daytime wind speeds at the surface we still measured higher $SO_2$ concentrations despite the remote locations of the $SO_2$ sources (the closest oil refineries around 40 to 70 km from the site).
Satellite measurements of $SO_2$ from TROPOMI (Fig. 11a) show that there are major sources further afield in the Arabian Gulf and in Iraq and Iran, with significantly elevated background values throughout the gulf itself and extending inland around the gulf. Figure 5 shows that winds are much stronger aloft than at the surface and that the wind direction is predominantly from the gulf, hence there is potential for long-range transport of $SO_2$ from the significant source areas seen in Fig. 11a if this $SO_2$ aloft is subsequently mixed down to the surface.
Combining the HALO Doppler lidar boundary layer evolution with our surface observations, we see increases in $SO_2$ once the boundary layer grows in height to begin mixing elevated layers down to the surface (Fig. 12) and that this increase is already seen by mid-morning, long before sea breezes are expected to reach the station. Figure 12 also shows that the majority of high

$SO_2$ concentration episodes (defined previously as being above the 90th percentile; 1.174 ppb) are when the wind direction at upper levels in the boundary layer is from the gulf.

To ensure that the observed wind directions measured aloft were representative of the larger meso-scale flow, air-mass back-trajectory analysis was was performed for all layers within the boundary layer and binned with respect to the surface $SO_2$ concentrations. The relative difference between airmass trajectories associated with high and low surface $SO_2$ concentrations measured at the station is shown in Fig. 11b and displays a very similar spatial pattern to the mean $SO_2$ concentration from the TROPOMI satellite (Fig. 11a); high relative difference values are consistently associated with regions having high mean

$SO_2$ concentrations while trajectories coming from e.g. the ocean (low mean satellite $SO_2$ concentrations) are in the lowest 10th percentile range. Thus, we have confidence that long-range transport is responsible for the elevated $SO_2$ concentrations measured at the station.

Our hypothesis is shown in Fig. 13, a schematic of different horizontal transport cases of $SO_2$. Example case a) is when there is horizontal $SO_2$ transport in both the upper and lower parts of an air column. In case b) we have horizontal transport from a

direction where there are no $SO_2$ sources. In the third example c) there is horizontal $SO_2$ transport only in the upper part of an air column and the transported $SO_2$ is mixed down to the surface when the boundary layer evolves and grows to the height of the air mass including $SO_2$. We consider example case c) as an explanation for the measured higher $SO_2$ concentrations at the surface during daytime when the surface winds speeds are low.

## 4    Conclusions

We used data from a one-year measurement campaign conducted in the United Arab Emirates during 2018–2019 to investigate how aerosol particle and $SO_2$ concentrations at the surface responded to changes in horizontal and vertical transport. In this sense, the measurement location was ideal, as the boundary layer structure was very similar from day to day, with a deep well-mixed boundary layer during the day, and a shallow nocturnal boundary layer. The maximum boundary layer height was usually reached in the afternoon around 1400 local time.

The potential for horizontal transport at different altitudes was obtained from the vertical profile of the horizontal winds measured by the Doppler lidar scans, together with the surface measurements. At this location, vertical transport is dominated by the turbulent mixing within the daytime convective well-mixed boundary layer, and the extent of the vertical transport was diagnosed from the Doppler lidar vertical-pointing observations. It is assumed that the vertical mixing timescale in the well-mixed boundary layer is of the order of 10 minutes. For the surface measurements, the horizontal transport can then be defined

with respect to the maximum wind speed in the vertical profile within the well-mixed boundary layer, as the vertical transport timescale will rapidly mix atmospheric constituents to all altitudes within the well-mixed boundary layer.

The measurement location was a rural background site, with local sources of $SO_2$ expected to be minimal. Our results indicate that the majority of $SO_2$ measured at the site has been transported from the major sources of $SO_2$ in the region, located in the gulf and further afield in Iran and Iraq. This was clearly seen in the response of the $SO_2$ concentration with the

wind speed and direction within the boundary layer as it evolves over time, with elevated concentrations seen for airmasses that passed over the sources in the gulf before arriving at the station.

The nearest cities and refineries are 40 - 100 km in distance from the measurement location. Surface wind speeds at the station at night were often less than $1\,\mathrm{m\,s^{-1}}$ which would equate to horizontal transport of 40 km taking over 11 hours (over 27 hours for 100 km). Together with a shallow boundary layer, this implies that vertical and horizontal transport at night is quite limited. However, the wind speeds above the nocturnal boundary layer usually remained high at night, often reaching $10\,\mathrm{m\,s^{-1}}$ or more, implying transport times closer to 1-3 hours. Therefore, as soon as the morning boundary layer begins to grow and entrain air from above, the surface can now experience the impact of significant horizontal transport; the constituents of an entrained elevated layer will be mixed vertically throughout the boundary layer on the order of 10 minutes.

Elevated layers arriving with trajectories that passed over $SO_2$ sources identified from measurements by the TROPOMI satellite satellite usually contained significant quantities of $SO_2$ and surface-measured concentrations increased rapidly once these particular layers started to be entrained into the growing boundary layer, even if the surface wind direction was from a clean sector. These conditions also often displayed higher nucleation mode aerosol particle concentrations, i.e. new particle formation events occurring due to the increase in the gaseous precursor.

The CCN number concentration and activation fraction did not show a clear dependence on the wind direction. The CCN number concentration showed some dependence on wind speed, with slightly higher concentrations under weaker winds. These higher concentrations were attributed to weaker dilution of aerosol particles within a shallower boundary layer, with a shallower boundary layer linked to lower surface wind speeds. A similar dependence of $SO_2$ concentration on low surface wind speeds was not seen, which was likely due to the source of $SO_2$ being in the elevated air mass and being entrained down to the surface through vertical mixing.

A comprehensive picture of the processes affecting aerosol particle properties means that it is not only important to investigate what is happening at the surface but also diagnose the impact of the dynamic boundary layer on the potential for significant horizontal transport in elevated layers and subsequent mixing in the vertical.

*Data availability.* The data used in this study are available upon request.

*Author contributions.* JK, EOC, AH and EA conceptualised the original paper. JK, JB and HL were responsible for the measurements. JK, EOC and JB processed the data. JK, EOC, JB, MF A-MS performed the data analysis. JK wrote the paper. All authors were involved in the interpretation of the results and paper editing.

*Competing interests.* At least one of the (co-)authors is a member of the editorial board of Atmospheric Chemistry and Physics.

*Disclaimer.* Any opinions, findings and conclusions or recommendations expressed in this material are those of the author(s) and do not necessarily reflect the views of the National Center of Meteorology, Abu Dhabi, UAE, funder of the research.

*Acknowledgements.* This work was supported by the National Center of Meteorology, Abu Dhabi, UAE, under the UAE Research Program for Rain Enhancement Science and the Academy of Finland Flagship funding (grant no. 337552). The work of J. Kesti is funded by the Maj and Tor Nessling Foundation (Grant 202000254). The authors are grateful to Timo Anttila, Siddharth Tampi and Farah Abdi for providing on-site technical support.

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

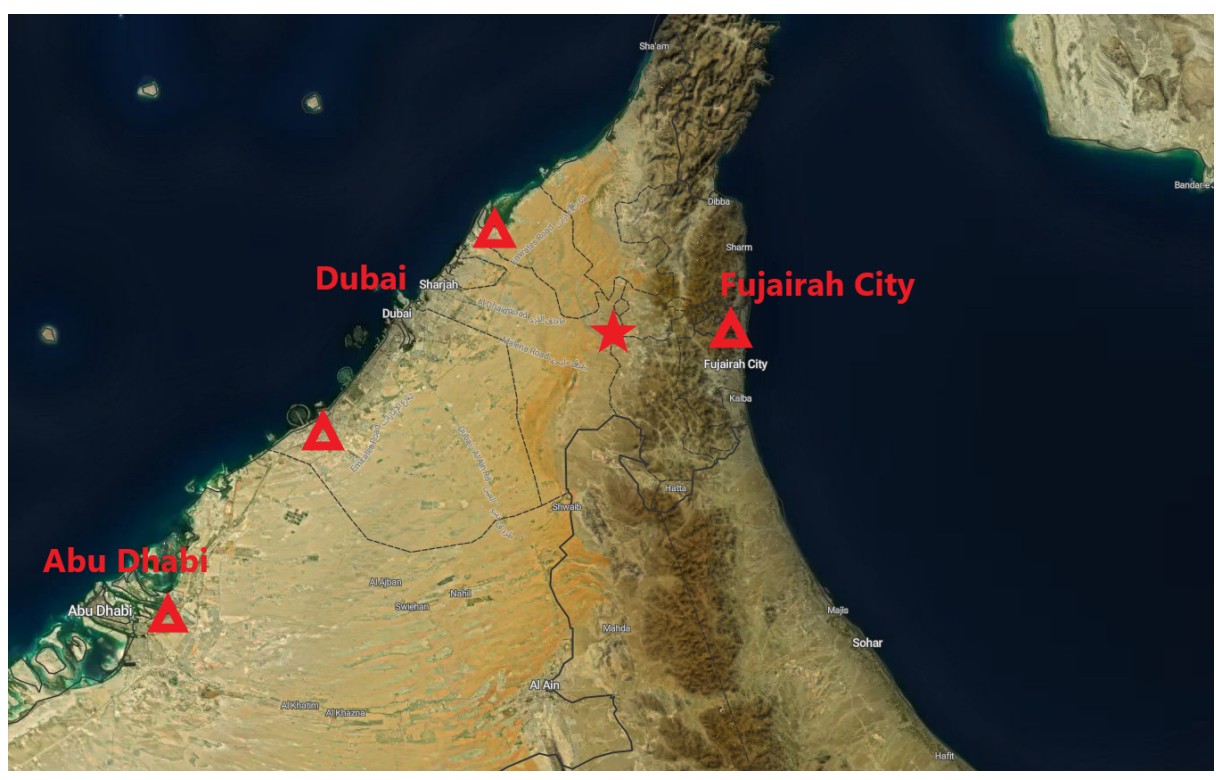

**Figure 1.** A map (Buchhorn et al., 2020) showing the location of the measurement site with a red star. The red triangles represent the locations of significant oil refineries known by the writers. © Copernicus Service Information 2021. Figure and caption from Kesti et al. (2022).

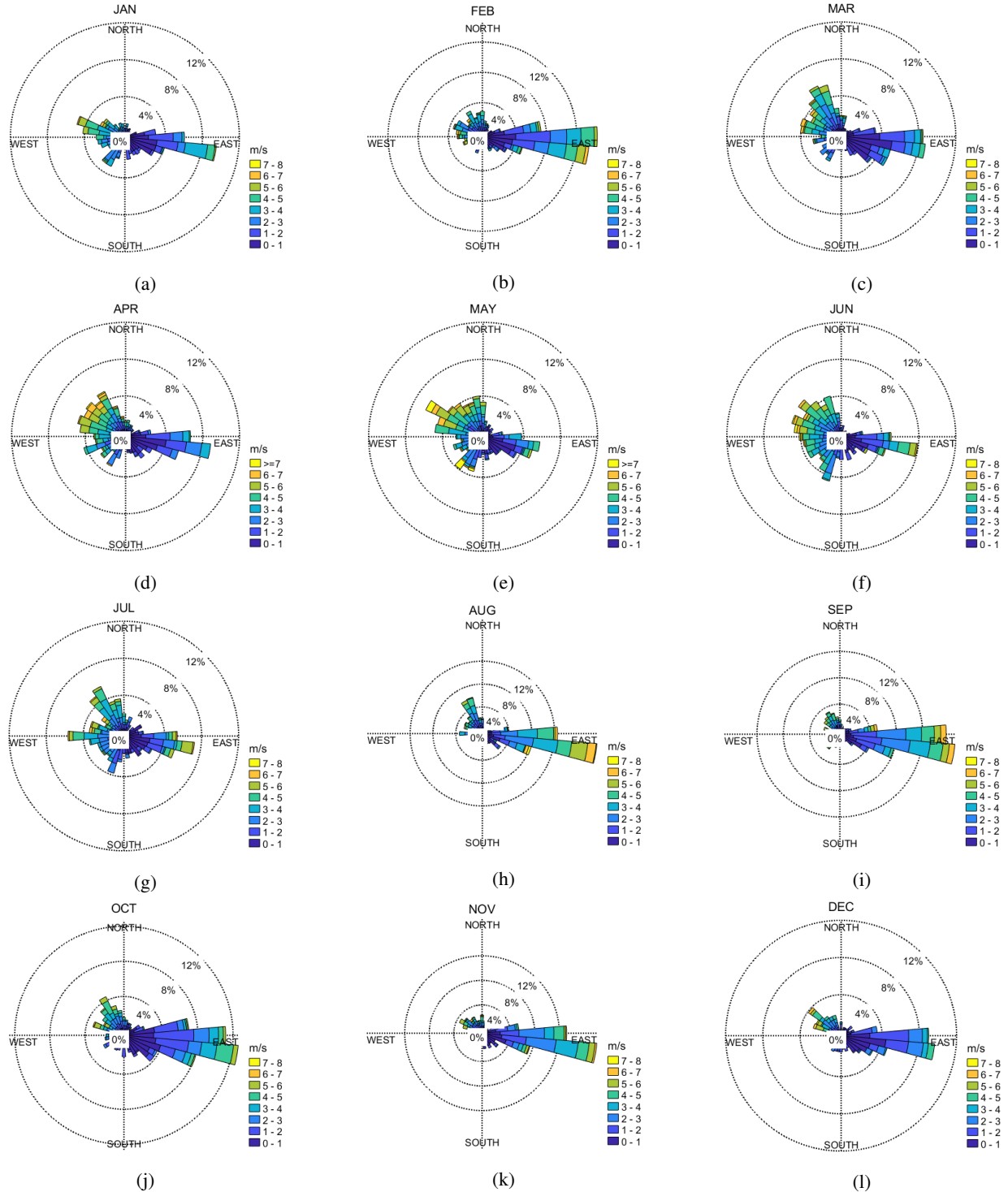

**Figure 2.** Wind roses measured close to the surface at the measurement site during the campaign, averaged over different months: a) January, b) February, c) March, d) April, e) May, f) June, g) July, h) August, i) September, j) October, k) November and l) December. The colours show the wind speed (m s$^{-1}$) and the percentages indicate the percentage of time when wind was observed from certain direction. The figure shows data from the automatic weather station.

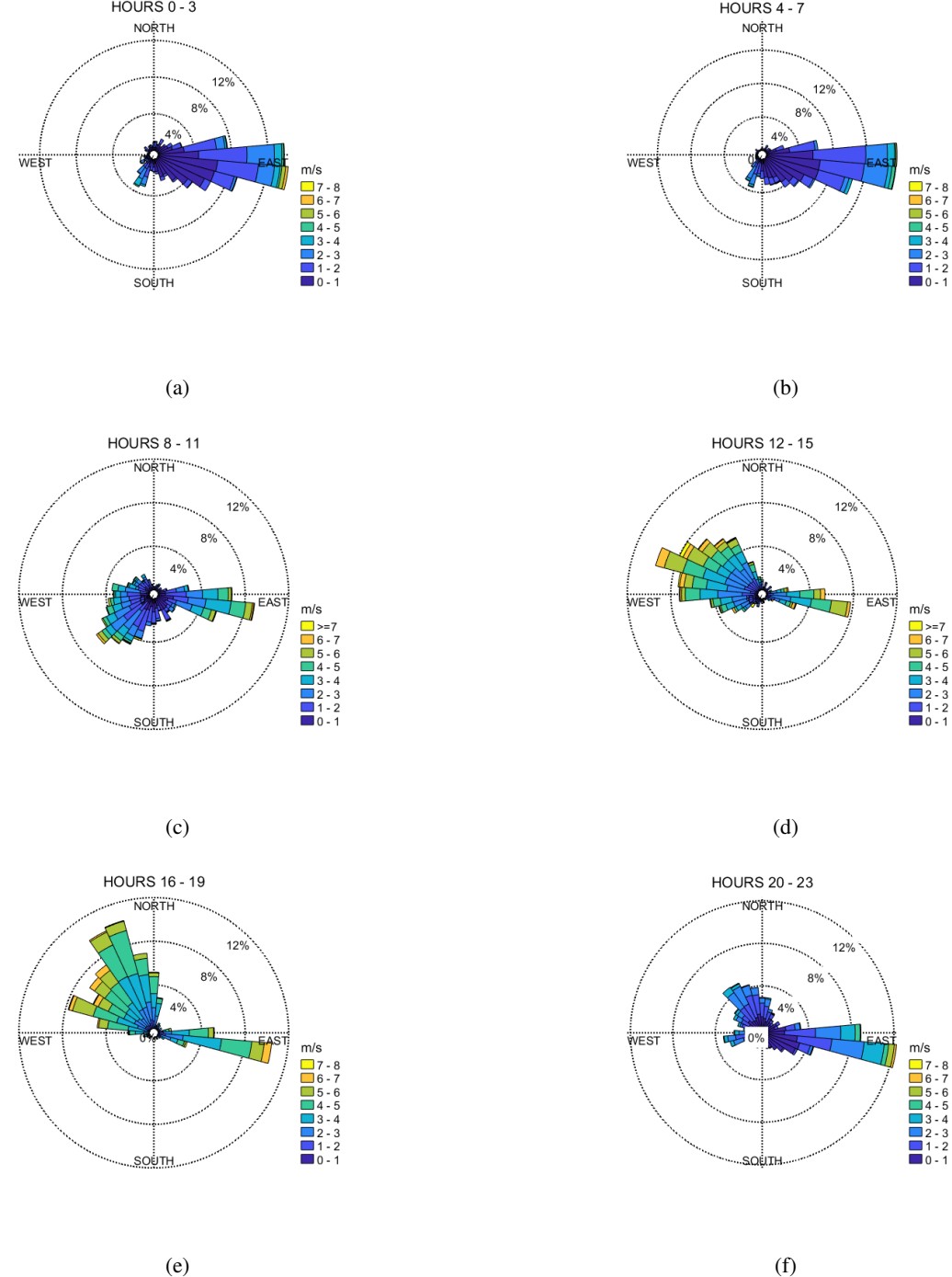

**Figure 3.** Wind roses measured close to the surface at the measurement site during the campaign, averaged over: a) hours 0–3, b) hours 4–7, c) hours 8–11, d) hours 12–15, e) hours 16–19 and f) hours 20–23, local time (hours UTC+4). The colours show the wind speed ($\mathrm{m\ s^{-1}}$) and the percentages indicate the percentage of time when wind was observed from certain direction. The figure shows data from the automatic weather station.

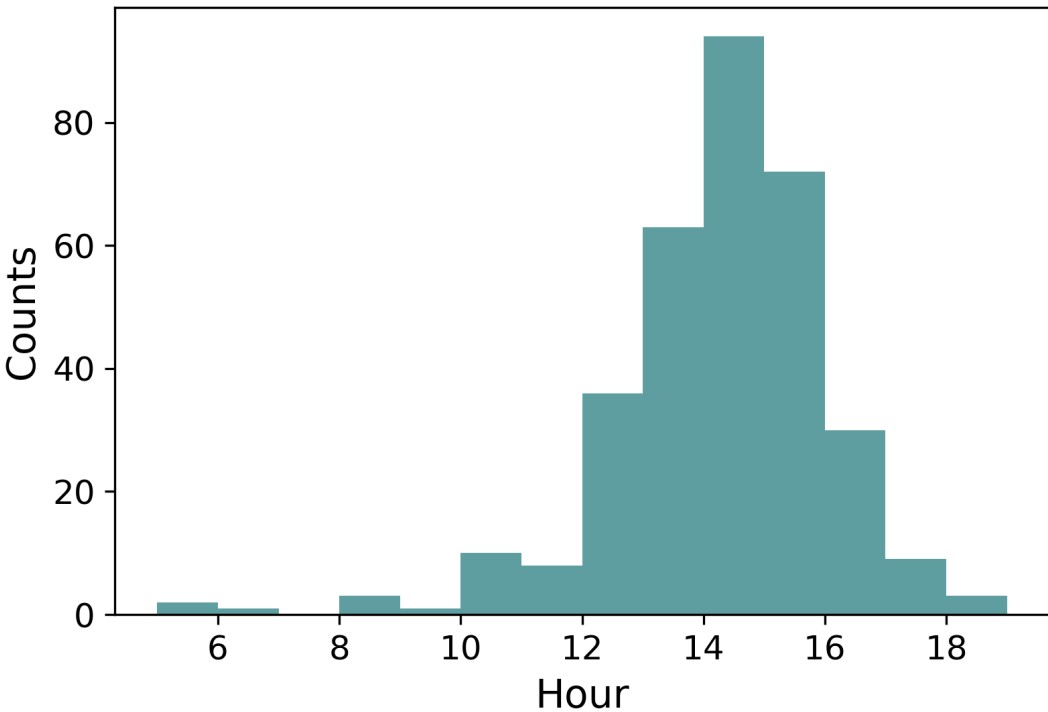

**Figure 4.** A histogram of hours when the boundary layer has reached its maximum height during each campaign day. The time axis is local time (hours UTC+4).

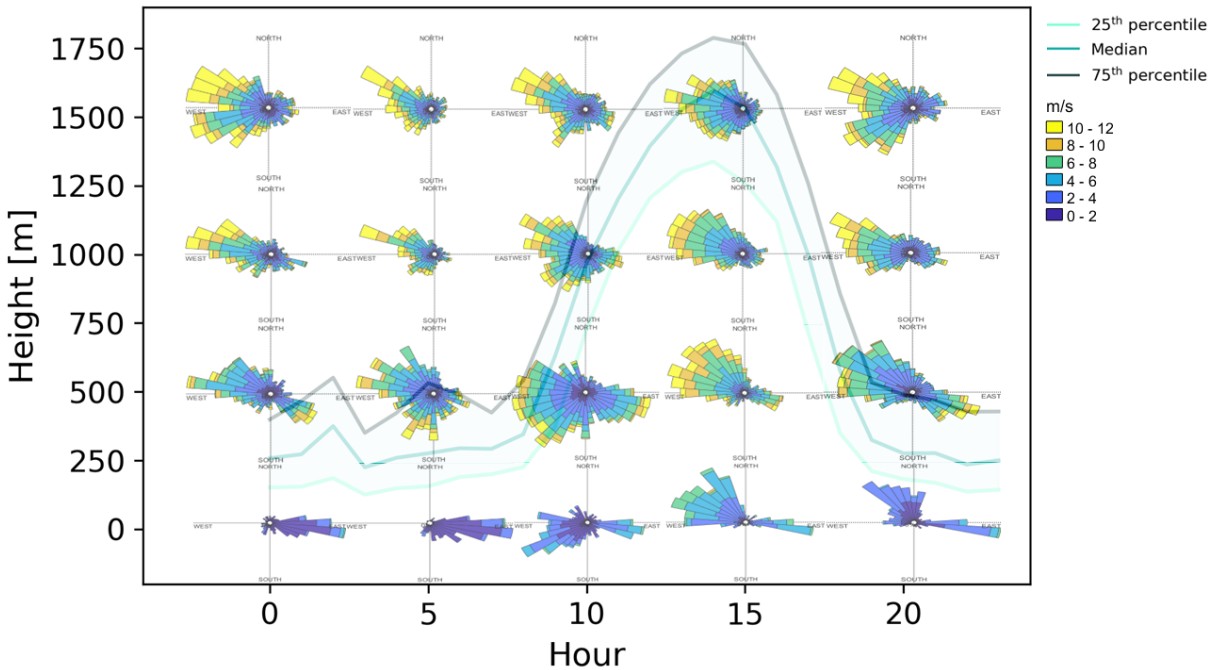

**Figure 5.** Wind roses at different heights: $1^{st}$ row from the bottom at surface, $2^{nd}$ row from the bottom at height 493 m, $3^{rd}$ row from the bottom at height 1000 m and $4^{th}$ row from the bottom at height 1508 m. The colours show the wind speed $(m\,s^{-1})$ and in the background figure the lines show the boundary layer height median with $25^{th}$ and $75^{th}$ percentiles. On the x-axis is the hour (local time UTC+4) and on the y-axis is the boundary layer height (m).

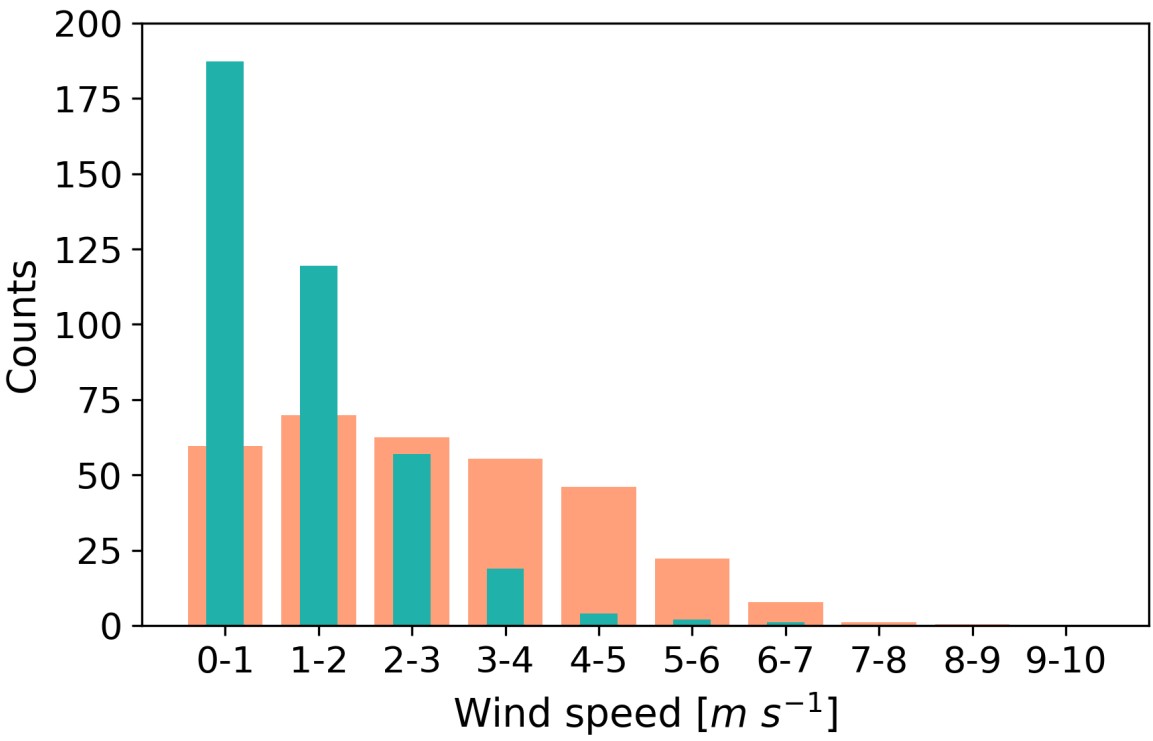

**Figure 6.** Histograms of surface wind speed during day (in orange) and night (green), showing counts per bin where the bin size is $1\ \mathrm{m\ s}^{-1}$ wide. The counts are normalised with respect to the different lengths of day (16 hours) and night (8 hours).

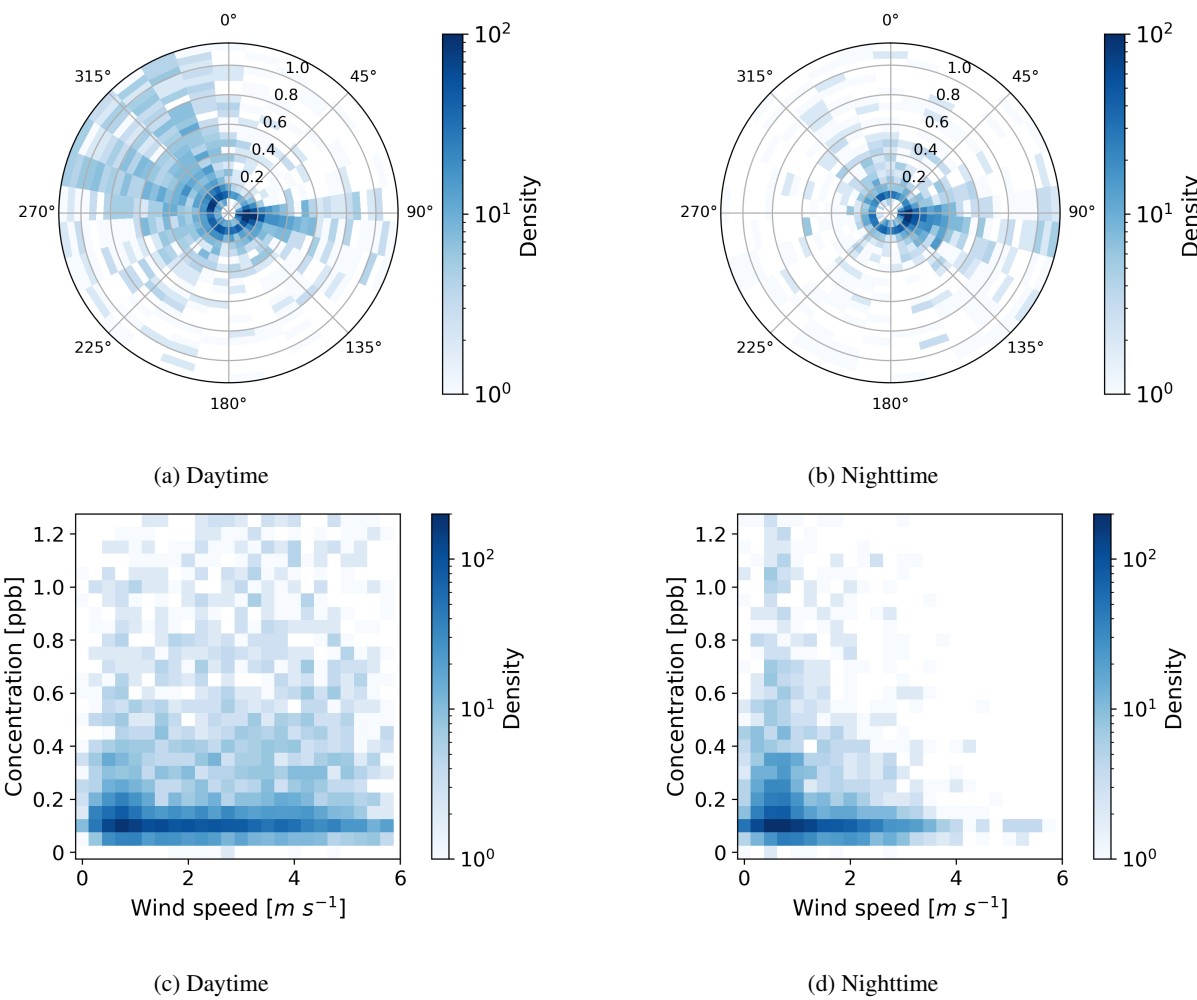

**Figure 7.** Polar plots of $SO_2$ concentration (ppb) with respect to wind direction (°) during a) daytime and b) nighttime. 2D-histogram plots of $SO_2$ concentration (ppb) versus wind speed (m s$^{-1}$) during c) daytime and d) nighttime. The colour indicates the amount of data points in the area and the radius of the circles in figures a) and b) indicate the $SO_2$ concentration in ppb.

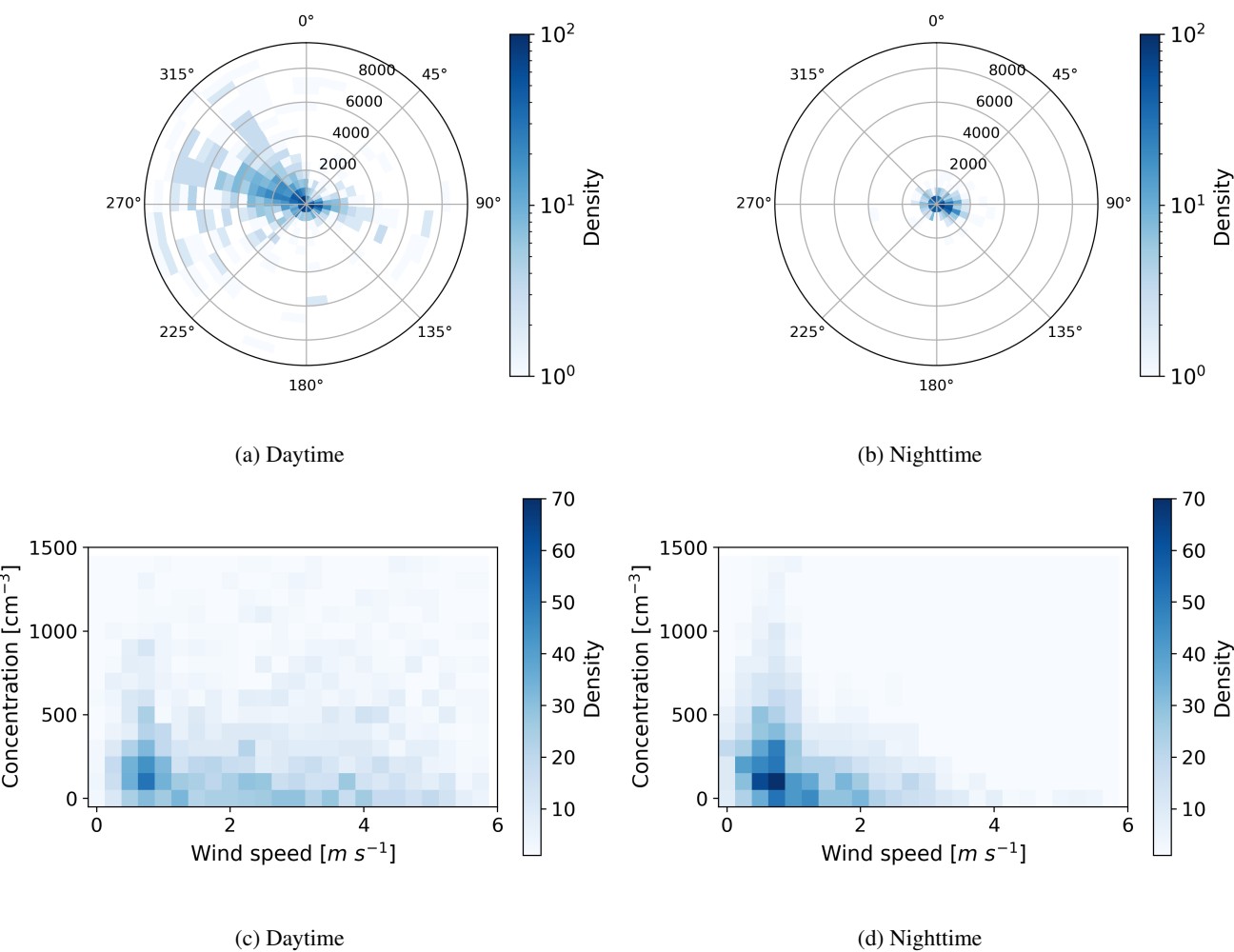

(a) Daytime

(b) Nighttime

(c) Daytime

(d) Nighttime

**Figure 8.** Polar plots of nucleation mode total number concentration $(\mathrm{cm}^{-3})$ with respect to the wind direction $(°)$ during a) daytime and b) nighttime. 2D-histogram plots of nucleation mode total number concentration $(\mathrm{cm}^{-3})$ versus wind speed $(\mathrm{m\,s}^{-1})$ during c) daytime and d) nighttime. The colour indicates the amount of data points in the area and the radius of the circles in figures a) and b) indicate the nucleation mode total number concentration in $\mathrm{cm}^{-3}$.

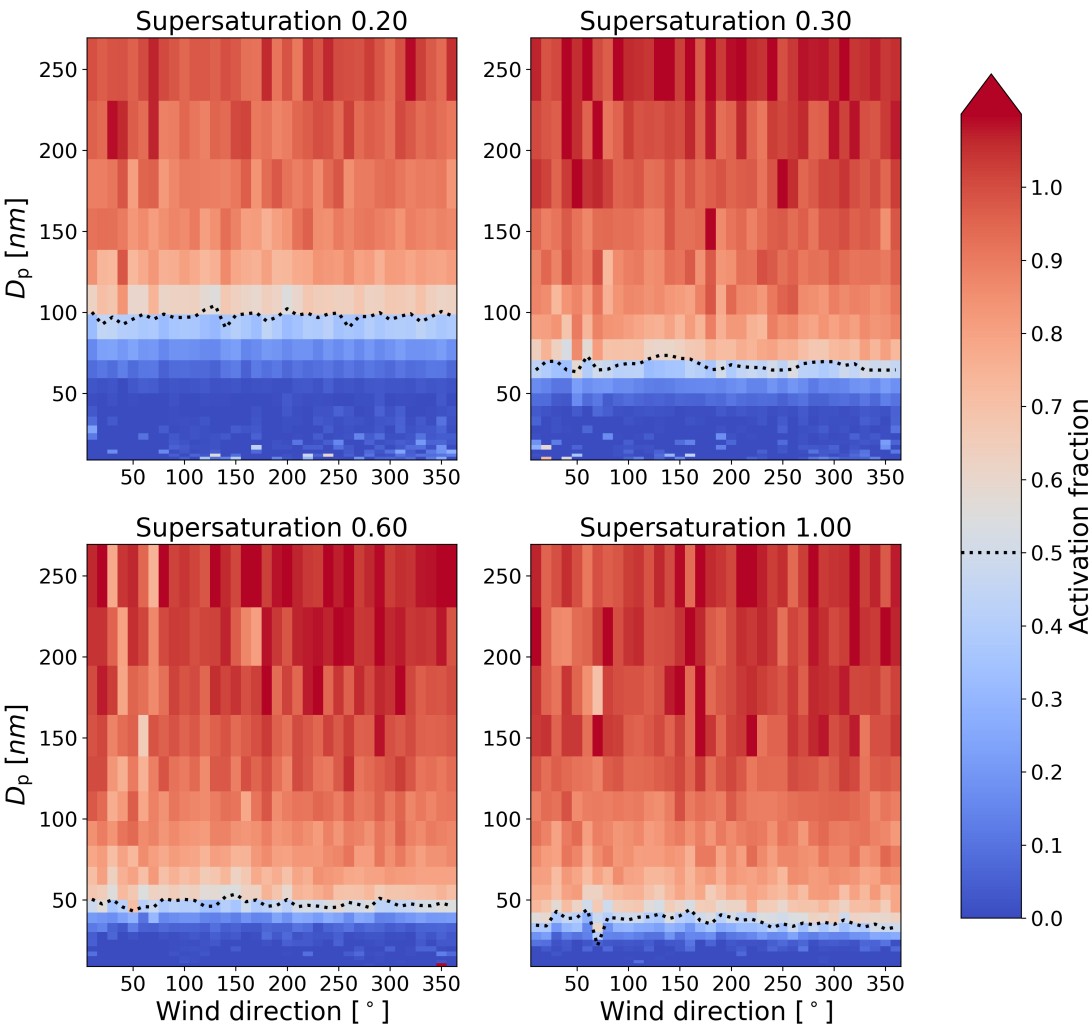

**Figure 9.** Particle diameter (nm) versus wind direction (°) at four different supersaturations (0.2, 0.3 0.6, 1.0). The colour indicates the activation fraction CCN/CN and the dotted line indicates the particle diameter at 50 % activation ($D_{50}$).

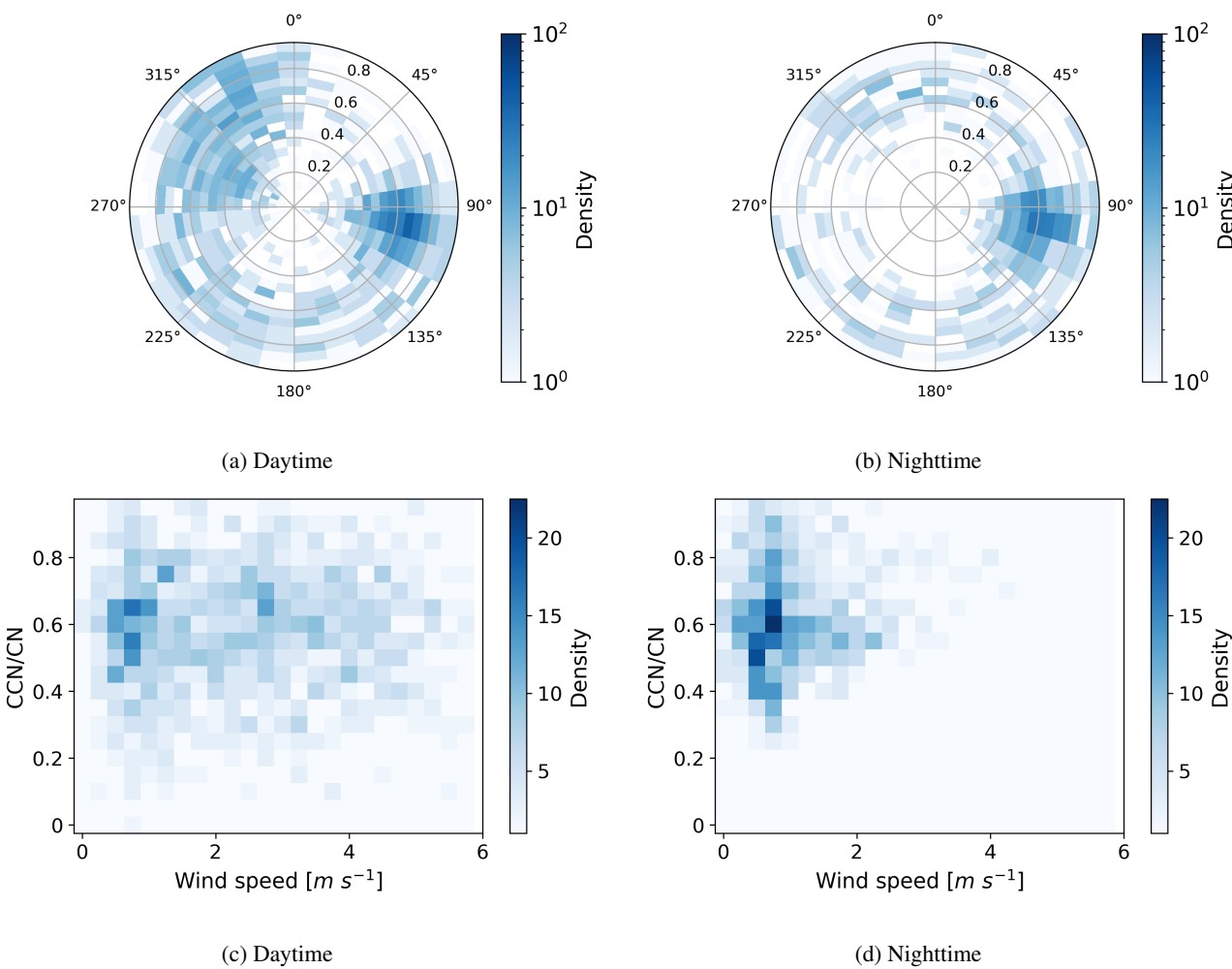

**Figure 10.** Polar plots of activation fraction (CCN/CN) at a supersaturation of 1.0 with respect to the wind direction (°) during a) daytime and b) nighttime. 2D-histogram plots of activation fraction (CCN/CN) at a supersaturation of 1.0 versus wind speed (m s⁻¹) during c) daytime and d) nighttime. The colour indicates the amount of data points in the area and the radius of the circles in figures a) and b) indicate the activation fraction at a supersaturation of 1.0.

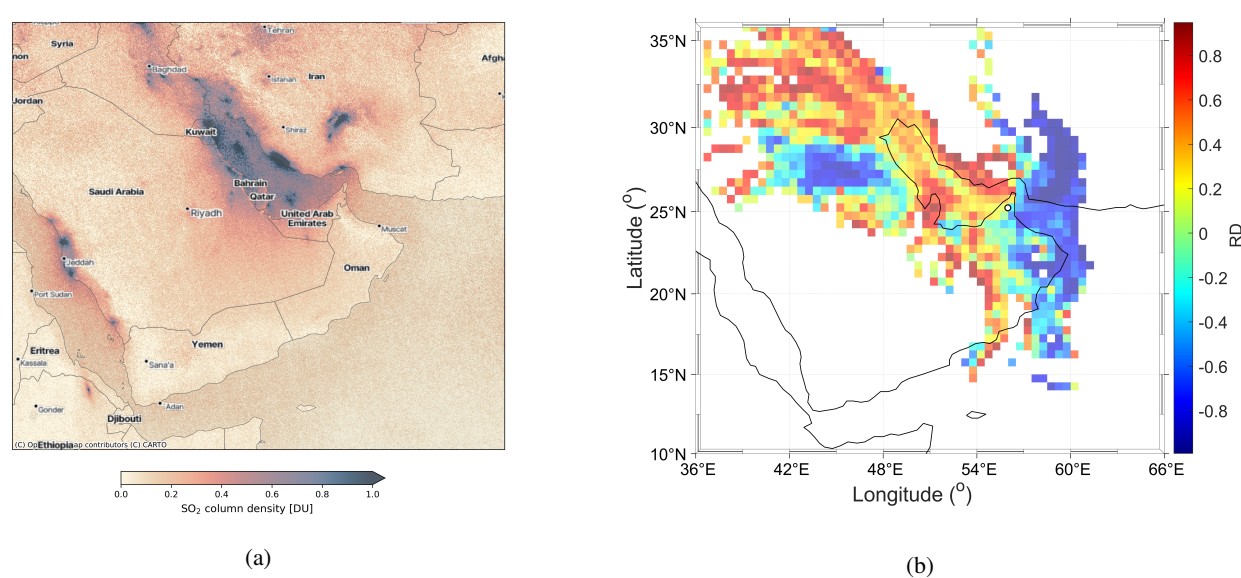

(a)                                                    (b)

**Figure 11.** (a) Mean $SO_2$ concentration from the TROPOMI satellite for the period from May 2018 until December 2019. The concentrations are shown in Dobson units (1 Du = 2.69 x $10^{16}$ molecules $cm^{-2}$). (b) Relative difference (RD) of the normalised counts between airmasses associated with high (H) $SO_2$ concentrations and low (L) $SO_2$ concentrations binned onto a 0.5 x 0.5° grid. High $SO_2$ concentrations denote values above the 90th percentile (1.174 ppb) and low concentrations below the 10th percentile (0.102 ppb). The RD is then calculated as RD = (H-L)/(H+L) for every grid cell that the trajectories have passed through. A maximum of 48-h air mass back-trajectories is shown.

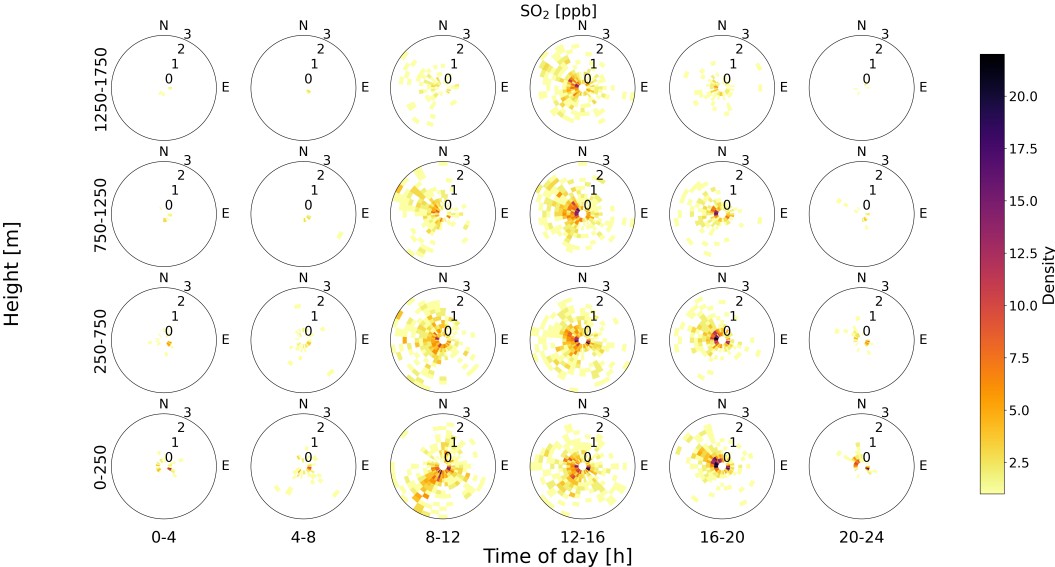

**Figure 12.** Polar plots of $SO_2$ concentration in ppb as a function of height (within the boundary layer) and time of day (local time, hours UTC+4). The rose plots for different heights have been constructed using HALO lidar data (wind speed and wind direction) at different heights in combination with $SO_2$ concentrations as measured at ground level. Only HALO lidar data from within the boundary layer is used in the figure.

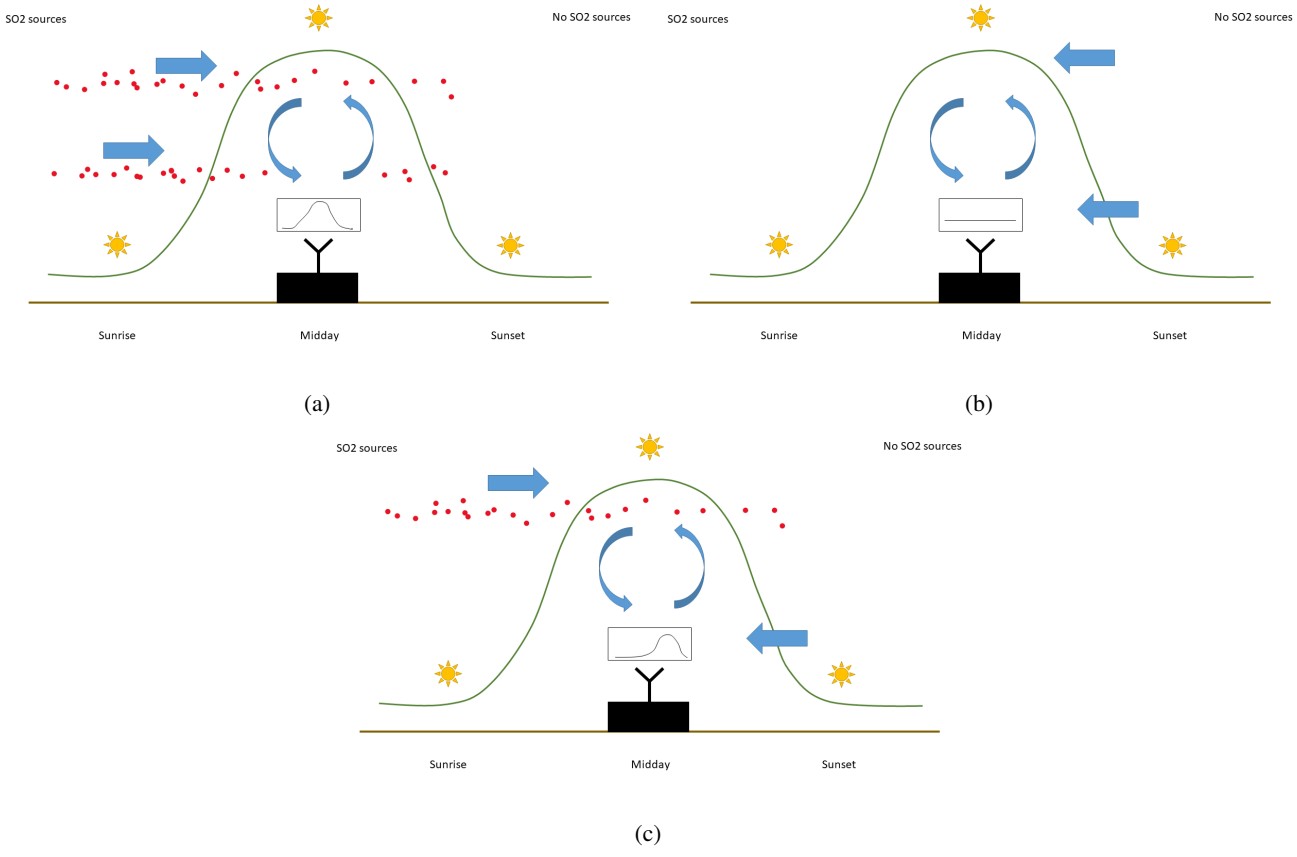

**Figure 13.** Schematic of $SO_2$ mixing in a boundary layer: a) $SO_2$ transport in the upper and lower parts of an air column, b) No $SO_2$ transport in an air column, and c) $SO_2$ transport in the upper part of an air column.

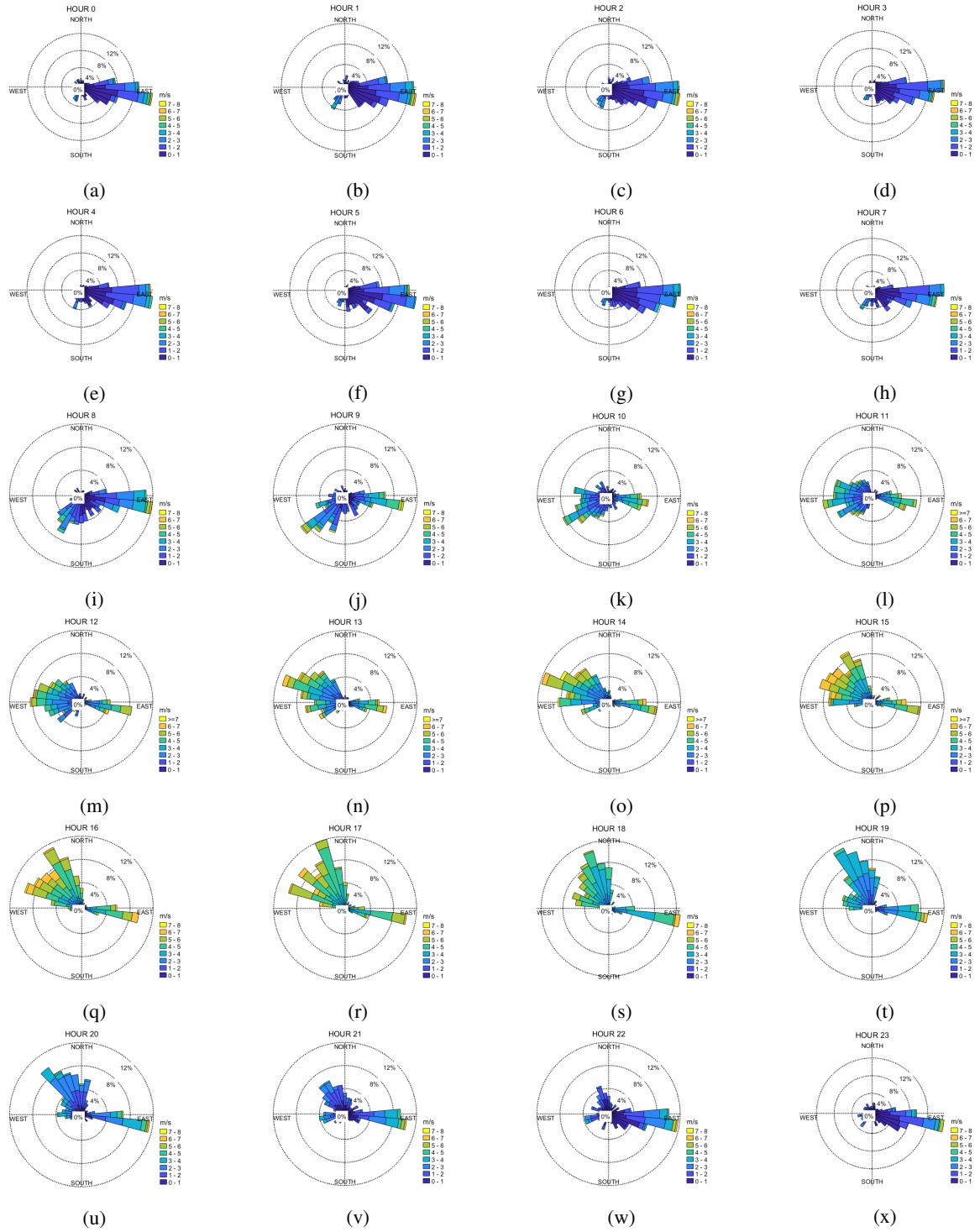

**Figure A1.** Wind roses for different hours in the UAE: a) 0, b) 1, c) 2, d) 3, e) 4, f) 5, g) 6, h) 7, i) 8, j) 9, k) 10, l) 11, m) 12, n) 13, o) 14, p) 15, q) 16, r) 17, s) 18, t) 19, u) 20, v) 21, w) 22 and x) 23. The colours show the wind speed $(\mathrm{m\,s^{-1}})$ and the percentages indicate the percentage of time when wind was observed from certain direction.

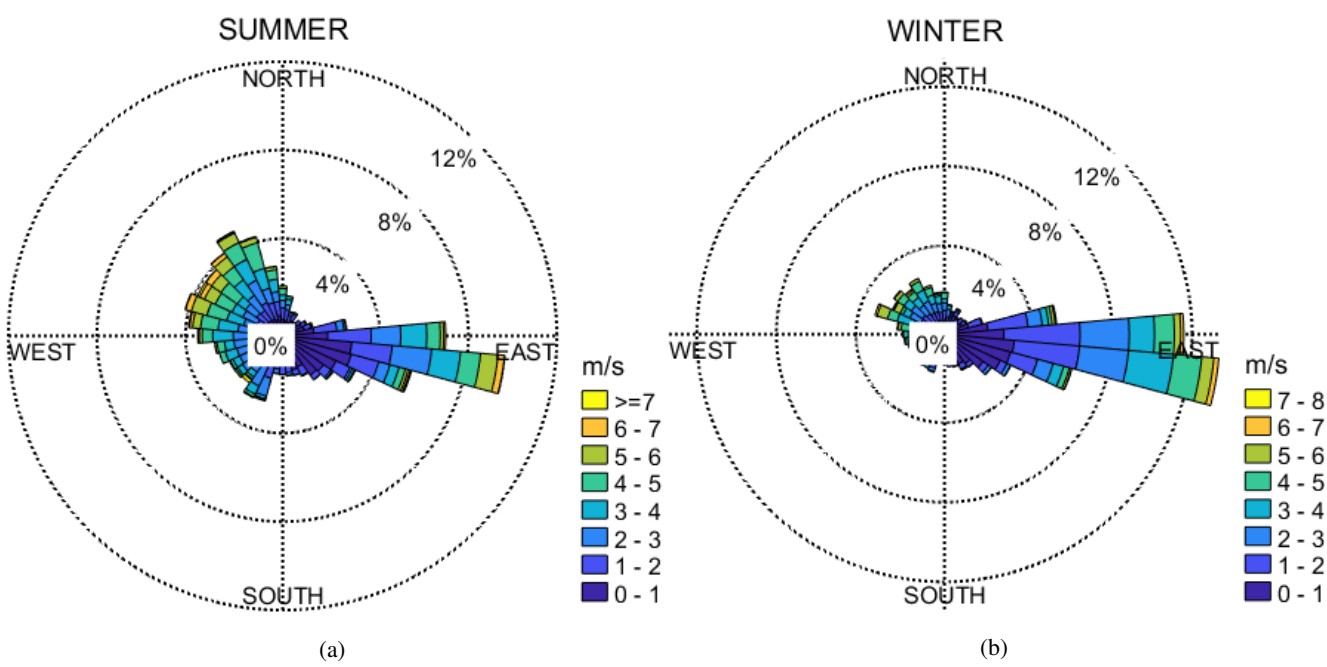

**Figure A2.** Wind roses for different seasons in the UAE: a) summer and b) winter. The colours show the wind speed (m s$^{-1}$) and the percentages indicate the percentage of time when wind was observed from certain direction.

**Table A1.** Mean, standard deviation, 25-, 50- and 75 percentile, minimum and maximum values for activation fraction and $\kappa$ values at four different supersaturations.

| Variable | Mean $\pm$ std | P $25^{\text{th}}$ | P $50^{\text{th}}$ | P $75^{\text{th}}$ |
|---|---|---|---|---|
| $\text{CCN/CN}_{\text{SS } 0.2}$ | $0.29 \pm 0.17$ | 0.16 | 0.27 | 0.39 |
| $\text{CCN/CN}_{\text{SS } 0.3}$ | $0.45 \pm 0.21$ | 0.30 | 0.44 | 0.60 |
| $\text{CCN/CN}_{\text{SS } 0.6}$ | $0.57 \pm 0.22$ | 0.43 | 0.56 | 0.71 |
| $\text{CCN/CN}_{\text{SS } 1.0}$ | $0.63 \pm 0.21$ | 0.50 | 0.63 | 0.77 |
| $\kappa_{\text{SS } 0.2}$ | $0.37 \pm 0.11$ | 0.30 | 0.35 | 0.43 |
| $\kappa_{\text{SS } 0.3}$ | $0.52 \pm 0.15$ | 0.41 | 0.50 | 0.61 |
| $\kappa_{\text{SS } 0.6}$ | $0.37 \pm 0.15$ | 0.27 | 0.36 | 0.47 |
| $\kappa_{\text{SS } 1.0}$ | $0.28 \pm 0.14$ | 0.17 | 0.26 | 0.37 |