# Peer review of "How horizontal transport and turbulent mixing impacts aerosol particle and precursor concentrations at a background site in the UAE"

_Atmospheric Chemistry and Physics, 2022_

## Author Comment (AC1)

Figure A: New figure replacing Fig. 9 in the previous manuscript version

[Figure]

Figure B: Dependence between CDNC and the standard deviation of vertical velocity with measured aerosol size distribution from 19th May, 2018 based on a box model calculation using parameterized cloud droplet activation by Abdul-Razzak and Ghan (2000, https://doi.org/10.1029/1999JD901161).

---

## Author Response (AR1)

**List of major and relevant changes to the revised manuscript**

- Four additional authors were added as co-authors to the manuscript due to the extent of the revisions and accompanying data analysis.

- A trajectory analysis was added to the manuscript as requested by the reviewers (including a figure Fig. 11b in the revised manuscript)

- TROPOMI $SO_2$ satellite data was included in the analysis and added as a figure (Fig. 11a in the revised manuscript).

- Figure 9 was removed and replaced by a figure showing the activation fraction for different super-saturations as a function of wind direction (Fig. 9 in the revised manuscript). Section 3.3.2 was also rewritten to a great extent.

- Section 3.3.3 was largely rewritten based on the reviewers comments and the new data analysis that they requested.

- A figure was added (Fig. 12 in the revised manuscript) showing the concentration of SO2 as a function of wind direction and height combining boundary layer height, wind direction from the HALO lidar and surface observations of $SO_2$.

- Figure 3 was moved to the appendix and replaced by a much less busy figure, as requested by the reviewers. The new figure shows the same information with less panels/subplots.

**Rely to Reviewer 1**

We thank both reviewers for their constructive comments. We have addressed all of the points raised by the reviewers (copied here and shown in black text), and include our responses to each point below (in blue text). Where there has been a major change in the manuscript we provide the original text (in black italics) and the new text (in bold + italics).

We have substantially improved the manuscript and have been able to address all the comments by both reviewers. The revised manuscript includes more in-depth analysis of how the meteorology affects the surface observations at the site. We have added satellite retrievals of $SO_2$, back-trajectory analysis, and profile observations with the HALO lidar combined with surface observations of $SO_2$. The revised manuscript now strongly supports the schematics shown as Fig. 11 in the original manuscript.

**Comment:** This work is a companion to the publication of Kesti et al. (2022). It summarizes the wind patterns of a background site in UAE and tries to link CCN, $SO_2$ and nucleation mode particles with the observed patterns to explain the variations of these parameters. The paper is fluently written but does not contain enough interesting results. These have already been reported by Kesti et al. (2022). The manuscript gives the impression of being the supplement of the first publication, with some of the data found in this work being already being presented.

The authors do a very good job in describing the meso/microscale wind circulation in the area. The problems arise when they try to link that with some measured variables.

**Reply:** We thank the reviewer for their constructive feedback. As the reviewer haspointed out, the manuscript partly overlaps with Kesti et al. (2022) and is, in some ways, a continuation of that article. To ensure that this manuscript is more than a supplement to the original article, we have revised the manuscript to include satellite retrievals and back-trajectory modelling, together with a more in-depth analysis of the surface and profile observations. The additional results are discussed in more detail in the individual comments below.

**Comment:** According to the authors the manuscript is focused on how air masstransport and boundary layer mixing conditions affect the Cloud Condensation Nuclei (CCN) activity measured in the surface. However, the analysis of their findings is rather incomplete and/or important information lacks from the manuscript. While they mention (cf. section 2 of the manuscript) that the CCN counter was coupled to a DMA in order to provide size resolved CCN activation fractions at different monodisperse aerosol sizes, this information is nowhere to be found neither in the text nor in the relevant figures. Accounting for the fact that in the supersaturation of 1.0 most of the accumulation mode particles will activate into droplets and therefore counted by the CCNc, reporting and depicting just the CCN concentration under various wind conditions (i.e., velocity and direction) does not contribute in understanding the implications of meteorological

conditions, wind origin and path combined with the activity of different sources in the CCN activity. Note that when the number concentration of the accumulation mode particles increases, the CCN concentration is also expected to increase at the supersaturation of 1.0. Therefore figure 9 does not add anything to the discussion and it should be removed.

**Reply:** The reviewer is correct that both the accumulation mode number concentration and CCN number concentration at a supersaturation of 1.0 should follow closely. The figure was removed as requested. A figure showing the activation fraction for different supersaturations and DMA-selected particle size as a function of wind direction was added to the subsection, including interpretation of the results. The sub-section of 3.3.2, which the reviewer found problematic, was largely rewritten to focus on the activation fraction of CCN/CN for different wind directions. Furthermore, the authors now refrain from speculation, as advised further on.

**Commet:** Instead the authors should: i) Provide the activation fractions (i.e., ratio of CCN concentration over aerosol number concentration; similarly to figure 10) but for the different

sizes they studied or at least for some representative sizes of the modes that are commonly observed in the atmosphere (i.e., nucleation, Aitken, accumulation).

**Reply:** See reply above.

**Comment:** ii) Calculate the size resolved hygroscopicity of the sampled monodisperseaerosols (e.g., by calculating the hygroscopic parameter "kappa"; cf. Petters and Krenidenweis, 2007) for the different meteorological conditions.

**Reply:** Again, in the revised manuscript Fig. 9 was removed and replaced by a figure showing the activation fraction (CCN/CN) as a function of aerosol particle (Dp) size and wind direction, as requested (See Fig. A the supplement pdf). The revised manuscript also provides a summary table including the calculated κ parameter. This data is shown for the supersaturations of 0.2%, 0.3%, 0.6% and 1%. The sub-section (3.3.2 in the original manuscript) was rewritten based on the reviewer's comments to focus on the new activation fraction analysis and the calculated κ values.

We also added a table (Table A1) to the Appendix showing statistics for activation fractions and kappa values at different supersaturations.

[Figure]

**Comment:** he above would provide much more useful information and perhaps some insights on how the different meteorological conditions and boundary layer evolution affect the hygroscopicity/CCN activity of the studied aerosols. In addition, since the measured (i.e., by the CCNc) CCN activity (especially at the extreme supersaturation of 1.0) does not necessary

reflect the effects of these aerosols on cloud formation the authors could use a simplified model for calculating the potential cloud droplet number concentration (cf. (Ghan et al., 2011; Morales Betancourt and Nenes, 2014). For instance, Kalkavouras et al. (2017) observed that during New Particle Formation (NPF) events over the Aegean Sea (Greece) the activation fraction of the sampled aerosols increased dramatically, however their effect on droplet formation was much lower.

In addition to that, the sensitivity of the potential cloud droplet number concentrations on

the type (i.e., chemical composition/hygroscopicity), size distribution of the sampled aerosols and on the meteorological conditions can be quantified by employing the above mentioned (or similar) models, which would significantly add value to the manuscript.

**Reply:** According to our observations, the aerosol particle concentration is commonly on the order of 5000 particles per cc or even more, while the composition of the particles is mostly $SO_2$ and organics. Thus, the resulting potential CDNC very much depends on the cloud type and the associated updraft velocities.

Figure B (see supplement pdf) shows an example result from running a simple cloud activation parameterization by Abdul-Razzak and Ghan (2000, https://doi.org/10.1029/1999JD901161). The calculation is performed using a measured aerosol size distribution from 19th May, 2018, together with several distributions of vertical velocities (assumed Gaussian) with varying standard deviation, as shown on the x-axis. For shallow cumulus or stratocumulus clouds, the vertical velocity distribution can be represented by a Gaussian distribution with a standard deviation σw which would likely reside around the 1-2 m s−1 range at maximum, which with very high particle concentrations is able to activate just a modest fraction of the total particle population. This results in a CDNC of about 1000 cm−3.

However, in convective clouds (which occasionally occur during summer-time at the UAE) the up-draft speeds can easily reach several meters per second, which can result in double the amount of droplets. This fits well with the range reported for CCN concentrations in the manuscript. Similar conclusions can be drawn from the results in e.g. Tonttila et al. (2022, https://doi.org/10.1175/JAMC-D-21-0183.1) based on LES model experiments with a more explicit cloud microphysical description.

In the prevailing conditions at the UAE the clouds usually reside at rather high altitudes. This affects the extent to which the clouds are coupled with the surface conditions, and thus the extent to which the surface-based aerosol measurements would be representative of the particle population in contact with the clouds. Moreover, co–variability between the aerosol and the meteorological conditions potentially exists in the area, so it remains an open question whether the aerosol properties presented here are representative of any specific cloudy situation. With these considerations, a more robust view into how the measured aerosol would translate into cloud microphysical properties in different meteorological conditions would require a quite extensive modelling closure study, which is certainly outside the scope of this paper.

[Figure]

**Comment:** The authors show that when specific wind directions prevail, there is $SO_2$ transport to the receptor site. This happens to occur during day-time when, conventionally, NPF occurs in most parts of the world. The authors provide no evidence that the transported $SO_2$ is taking part or enhancing the NPF process. Therefore the discussion in lines 179-183 is highly speculative and should be removed. There is growing evidence that sulfuric acid alone cannot account for NPF formation rates and additional constituents are required. It has also been shown that acidic conditions may inhibit NPF (Pikridas et al., 2012, doi:10.1029/2012JD017570.) I understand that the authors try to link the three variables they are reporting, but this is not a valid way. The same holds for discussion in line 220.

**Reply:** The referee raises a good point. If there is a lot of $SO_2$ and no components to neutralize it, acidic conditions may inhibit NPF. Still, one of the most important new particle formation precursors in the atmosphere is sulfuric acid ($H_2SO_4$), which forms from $SO_2$. We agree that the phrasing could be improved and therefore we have modified the text in section 3.3.1 as

follows: *This result is consistent with the $SO_2$ concentration (Fig. 7a), which is **supporting the fact that one of the most important new particle formation precursors in the atmosphere is sulfuric acid (H2SO4), which forms from sulfur dioxide (Weber et al., 1997; Kulmala et al., 2000; Sipilä et al.,2010).*** References: Weber et al. https://doi.org/10.1029/96JD03656, Kulmala et al. https://doi.org/10.1038/35003550, Sipilä et al. https://doi.org/10.1126/science.1180315

**Comment:** The authors should really use polar plots, or any other bivariate plot, to pass their message through. Polar plots can replace figures 7-10, and 12.

**Reply:** We have changed figures 7a, 7b, 8a, 8b, 10a and 10b to polar plots. Fig. 9 and 12 were removed.

**Comment:** The authors needs to show a map of all the refineries in the area in combination with an elevation map. Satellite maps can be used to identify the refineries. This would assist in the discussion found in lines 172-175.

**Reply:** The referee raises a good point. In addition to the map, which shows the location of the mountains and also the refineries known by the writers, we have added a figure displaying satellite measurements of $SO_2$ concentration. The figure showing $SO_2$, as seen by the TROPOMI satellite, is shown in a subplot in the revised manuscript in section 3.3. The $SO_2$ subplot

shows the column density in Dobson units retrieved from the TROPOMI satellite measurements from May 2018 until February 2019. The other subplot shows the $SO_2$ based trajectory analysis with back-trajectories arriving to the site within the boundary layer. We have noted that when interpreting the satellite figure the reader has to take into account that satellite measurements are not directly comparable with the surface measurements because the satellite measures the whole air column.

**Comment:** Fig. 3 is too big and does not assist the reader. Can the authors put it in the supplement and replace it with a more concise figure. Eg by lumping several hours together with similar profiles.

**Reply:** We have replaced Figure 3 with a figure where four hours are combined in each wind rose. The original figure with hourly values has been moved to the Supplement. The following text has been added to the section 3.1: ***Wind roses for every hour separately can be seen in Fig. A1.***

**Comments on analysis.**

**Comment:** My view is that analysis presented in this work is very poor. It does not into deep on any front and leaves the reader with many questions. This is one of the big weaknesses of the manuscript.

**Reply:** The analysis in the revised manuscript has been substantially improved and extended, making use of back-trajectories, satellite retrievals, and coupling HALO lidar profiles to surface observations.

**Comment:** The authors should discuss how the seasonal variation in wind direction is linked to synoptic conditions in the area. A trajectory analysis is a must. It was also requested to Kesti et al. (2022), but did not materialize.

**Reply:** We have added extra discussion on the meteorological conditions in the area in section "3.1 Observed meteorological patterns" of the revised manuscript. We have also added a trajectory analysis as requested which further adds to the understanding of the origin of the air pollution at the site. We restricted the back-trajectory analysis to $SO_2$ and not all

measured pollutants, since $SO_2$ has a relative short atmospheric lifetime compared to e.g. aerosol

particles and should therefore be more related to relatively local atmospheric conditions than very long-range transport patterns, which would make the analysis unnecessarily convoluted.

**Comment:** When report high/low concentrations please also report absolute numbers along with an error metric (eg ±1std) throughout the manuscript.

**Reply:** This is a very good point. In the revised version of the manuscript we provide error metrics for all relevant variables and parameters.

**Comment:** The authors compare concentrations of $SO_2$ in Section 3.3.1, but fail to report on any metric. What is elevated $SO_2$ concentration, how much is low? Are these differences substantial? How would these bias a measurement done once per day, eg by TROPOMI? These are just some questions I would like to see answered.

**Reply:** We have added metrics to section 3.3.1 as described in the response to the previous comment. We now define High and Low $SO_2$ concentrations within the trajectory analysis (Low ($SO_2$ <= 0.1019 ppb) and High ($SO_2$ >= 1.1740 ppb) based on percentiles and note that these differ by a factor of 10. We suggest that it is not straightforward to directly compare surface and satellite $SO_2$ measurements since the satellite is measuring an integrated atmospheric slanted column, where the depth in altitude of the $SO_2$ is not known. Additionally, a satellite measurement at one

location is made once per day at best (depends on cloud cover) and then usually averaged to obtain sensitivity (monthly is typical, unless in exceptional situations such as volcanic eruptions). However, the elevated $SO_2$ region seen in the satellite figure are in good qualitative agreement with the regions identified by the trajectory analysis.

**Comment:** Are the nucleation mode particles discussed in 3.3.1 part of NPF or just the background concentration. Is there any difference if NPF is involved with respect to wind direction and speed?

**Reply:** Nucleation mode aerosol particles are formed through NPF or they are directly emitted from traffic (R̈onkk̈o et al. 2017, https://doi.org/10.1073/pnas.1700830114). The remote location of the measurement site from larger cities would suggest that NPF is the source of nucleation mode particles at this site. After nucleation mode particles have been formed, they grow to larger sizes and therefore there would not be any background concentration of the smallest particles. The effects of wind direction and speed on the nucleation mode aerosol particles and hence

NPF are shown in Fig. 8 and discussed in section 3.3.1. Figure 8 shows that the nucleation mode aerosol particle concentrations are elevated in eastern and western wind directions, with higher concentrations in the western airflow. In daytime there is no clear dependence of nucleation mode aerosol particle concentration on wind speed (Fig. 8c), and during nighttime (Fig. 8d) the distribution follows the shape of the nighttime wind speed distribution in Fig. 6.

**Comment:** he discussion in Section 3.3.2 is very problematic as already noted. Please see general comments above on how to improve. The discussion on figure 9 (strongly suggesting to be removed) does not add any valuable information. The rest of the discussion in section 3.3.2 is speculative and probably misleading, while - based on the available measurements – conclusions could be drawn by following a more detailed analysis (cf. general comments for suggestions).

**Reply:** Section 3.3.2 has been rewritten and the speculative parts were removed. In the revised manuscript, section 3.3.2 focuses on the critical diameter as a function of wind direction. An additional table summarising the critical diameters and κ values are included. Moreover, the revised manuscript includes a trajectory analysis and satellite retrievals to show the origin of

$SO_2$, supporting the conclusions. Figure 9 was removed.

**Comment:** Line 210. It is not clear why this explanation was chosen, even though I agree with the authors. A more detailed discussion is required.

**Reply:** In the revised manuscript we show much more evidence for our explanation. We have used surface $SO_2$ concentrations and plotted those against wind directions within the boundary layer as determined by the HALO lidar. The figure shows that during shallow boundary layer conditions (during most nights) $SO_2$ concentrations are low. When the boundary layer is deep (during most of the day) the concentrations are much higher and are mostly associated with winds from the west where satellite retrievals show that there are sources of $SO_2$.

**Rely to Reviewer 2**

We thank both reviewers for their constructive comments. We have addressed all of the points raised by the reviewers (copied here and shown in black text), and include our responses to each point below (in blue text). Where there has been a major change in the manuscript we provide the original text (in italics) and the new text (in bold + italics).

We have substantially improved the manuscript and have been able to address all the comments by both reviewers. The revised manuscript includes more in-depth analysis of how the meteorology affects the surface observations at the site. We have added satellite retrievals of $SO_2$, back-trajectory analysis, and profile observations with the HALO lidar combined with surface observations of $SO_2$. The revised manuscript now strongly supports the schematics shown as Fig. 11 in the original manuscript.

**General comments**

**Comment:** The manuscript analyses wind data measured with a Doppler lidar together with CCN-, $SO_2$- and nucleation mode particles concentrations. The aim of the study is the change of concentrations measured near the surface and explain these changes by horizontal and vertical transport. The authors constraint the analysis their own data during a one-year measurement campaign. The manuscript seems to be a follow-up of the paper by Kesti et al. 2022 where already a seasonal analysis and some cases studies were presented. In this manuscript the authors wanted to tackle the boundary layer and transport more broadly which shows only limited novelty.

**Reply:** We thank the reviewer for the constructive feedback which helped us to improve the manuscript. The revised manuscript has been substantially improved based on both reviewers' comments. Both reviewers found the manuscript to have limited novelty when viewed in light of

what was already published by Kesti et al. 2022. In response, we have substantially extended the data analysis in the revised version of the manuscript. We have included a trajectory analysis, $SO_2$ satellite retrieval, and a more in-depth analysis of surface concentrations and transport of pollutants with the vertical profiles from the HALO Doppler lidar.

**Comment:** The authors aimed in describing and analysing the atmospheric transport of atmospheric constituents to a rural site in the UAE. And they tried it based on measurements of a single site (palm tree farm) and on few known sources of $SO_2$ without using additional data or model simulations. For a comprehensive picture this is not enough. Many questions remain, especially how well the local wind directions are representative for the transport pattern.

**Reply:** In the revised manuscript we have included a back-trajectory analysis combined with the vertical observations of the mixing layer depth and wind profile to show how the surface observations are connected to the boundary layer evolution and boundary layer depth. Air mass back-trajectories were computed every hour, every 200 m in altitude from the surface to the top of the mixing layer at that hour. Then, these trajectories were collocated in time with $SO_2$ observations and further separated into Low ($SO_2$ <= 0.1019 ppb) and High ($SO_2$ >= 1.1740 ppb) concentrations utilising the 10-90th percentile ratio from the $SO_2$ observations.

The trajectories were further binned onto a horizontal 0.5 x 0.5° grid keeping the counts passing through each grid cell to further derive the relative difference (RD) expressed as RD = (H-L)/(H+L) for every grid cell that the trajectories have passed over.

The revised manuscript also includes satellite measurements of $SO_2$ from TROPOMI for the region which show that the major sources are in the Gulf, in Iraq or in Iran, and that there are a few local sources on the Arabian peninsula (Fig. 1 in the submitted manuscript). There are no significant sources of $SO_2$ to the east of the site. We get a similar picture from the relative difference plot generated from the back-trajectory analysis showing that air masses passing over the vicinity of the sources seen in the satellite measurements are associated with high $SO_2$ concentrations, and those from the ocean are associated with low concentrations.

**Comment:** I am missing a discussion about sources and sinks. For instance: How big is the background? Could satellite data contribute to the identification of the $SO_2$ sources? Is the $SO_2$ which comes from the oxidation of DMS in the ocean relevant to the observed $SO_2$ level at the measurement site?

**Reply:** We now include satellite measurements of $SO_2$ in the revised manuscript along with a trajectory analysis. The satellite figure clearly shows major sources of $SO_2$ in the Gulf and in Iraq and Iran, with significantly elevated background values throughout the Gulf itself and extending inland around the Gulf. The satellite retrieval suggests that there is no significant source of $SO_2$ from the ocean, at least not relative to the sources from the Gulf. A similar spatial pattern is shown in the trajectory analysis, where trajectories coming from the ocean are in the lowest 10th percentile range. This is quite clear in the trajectory relative difference figure, which highlights the contrast in trajectories from the ocean with those passing over the Gulf. This does not mean that the ocean is not a source of $SO_2$, but that sources from the Gulf dominate any others.

Cloud formation and subsequent wet deposition is a major sink of $SO_2$ globally, but inland, at our measurement site at least, boundary layer clouds and precipitation were uncommon. We refrain from speculating too much about sinks and distant sources of DMS and their impact on our measurements; the other reviewer thought that some parts of the original manuscript were too speculative and should not be included.

**Comment:** I am also missing that the discussion of the transport include land-sea-breezes. Eager et al. (2008, JGR) used a network of surface stations in the United Arab Emirates (UAE). They found that a sea breeze occurs during all seasons of the year and that the horizontal extend onshore can be a large as 80km. Hence the palm tree farm is well within reach of the sea breeze.

**Reply:** Section 3 has been revised to include relevant references and discuss the implications of the sea breeze. However, the sea breeze is usually not observed at the coast until midday local time and is not observed 15-20 km inland until about 1500 local time according to Eager et al. (2008). Wind changes resembling a sea breeze were seen at our station but later in the afternoon when the boundary layer was already deep. Our HALO Doppler lidar wind profiles show that there is significant transport aloft (from the direction of the Gulf) throughout the early morning and during the day, which corresponds with the larger mesoscale wind direction (north-westerly flow) for the region which seems to be persistent all year round (Eager et al, 2008; Zhu and Atkinson, 2004, http://dx.doi.org/10.1002/joc.1045).

Our surface observations show that we see increases in $SO_2$ once the boundary layer grows in height to begin mixing these elevated layers down to the surface and that this happens in mid-morning, long before a sea breeze would reach the station. This does not preclude sea breezes also being a transport source, but that they do not seem to be the dominant source for $SO_2$.

**Comment:** They authors did a good job in describing different transport regime together with vertical mixing (e.g. figure 11). However, they did not analyse sufficiently the data with the aim of supporting the transport hypotheses. Why the sector analysing is just applied to surface data and the vertical change of the wind direction is not considered? Figure 5 nicely shows how the wind changes with altitude.

**Reply:** These shortcomings are rectified in the revised manuscript. As described above, we now use a back trajectory analysis approach which combines a set of back-trajectories corresponding to the depth of the observed mixing layer at each hour, and the surface $SO_2$ observations, in order to identify transport routes. This methodology permits the combination of trajectories potentially arriving at the site from different directions at different altitudes and simplifies the presentation.

**Comment:** There is also some doubts on the definition and naming of the sectors. Only four sectors are defined for the analysis although the wind roses exhibit much more sectors. However, under the assumption that the major refineries are the significant point sources for the measured $SO_2$ levels it is expected that a refined sector analysis (more smaller sectors) could provide a clearer picture of the transport.

**Reply:** The transport sector analysis has been rectified with a methodology using back-trajectories as described above. Together with the satellite measurement $SO_2$, the methodology provides a much more robust interpretation of our results.

**Comment:** Figure 12 does not show clear differences between so-called clean and polluted sectors. The $SO_2$ concentration values cover in both cases the range from 0.1 to 10 ppb. Only very few data points are larger for the polluted cases compared to the clean case. It's similar for the nucleation mode concentration. Most data points – although less compared to $SO_2$ for both sectors are overlapping. It seems that the figure underlies the analysis of the wind direction independently from the pollution level. And hence the names of the sectors – clean and polluted – is confusing.

**Reply:** The transport sector analysis has been rectified with a more appropriate methodology as described above. We removed the original Fig. 12 from the manuscript. We then added a new figure, comprising polar plots of $SO_2$ concentration in ppb as a function of height and time of the day. The rose plots for different heights have been constructed using HALO lidar data (wind speed and wind direction) at different heights in combination with $SO_2$ concentrations as measured at ground level. Only HALO lidar winds from within the boundary layer is used in the figure.

**Comment:** Furthermore I wonder whether wind direction data are reliable or can be used for transport analysis when the wind speed is very low. Calm conditions were excluded from the analysis?

**Reply:** It is correct to state that the wind direction is uncertain when the wind speed is very low. Many national meteorological weather centres do not show a wind direction in their online services when wind speeds are below $1 - 2$ ms$^{-1}$ (often replacing an arrow with a circle). However, in terms of long-range transport analysis, calm conditions do not necessarily need to be excluded as they will indicate local-only conditions when averaged to a horizontal grid. This is particularly true when using back-trajectory analyses averaged to a 0.5 x 0.5° grid for example, any wind direction would still result in calm conditions being averaged to the station pixel even over many hours.

**Specific comments**

**Comment:** Line 213/214: the division of the data into sectors is based on the Vaisala weather station wind data or based on the Halo Doppler Lidar data near the surface? I assume it's the weather station but I am not sure.

**Reply:** The transport sector analysis has been rectified with a more appropriate methodology as described above. HALO Doppler lidar wind data are used.

**Comment:** Lines 163 ff and fig. 6: Although the distribution of the wind speed does not change, a normalization of the counts with respect the different length of day (5-20 LT = 16 hours) and night (21-4 LT = 8 hours) is recommended.

**Reply:** The referee raises a very good point. We have normalised the counts with respect to the different lengths of day and night in Fig. 6.

**Minor remarks**

**Comment:** Figure 1 and caption: as it is re-used from Kesti et al. (2022) in my opinion the authors should write "figure and caption from Kesti et al." instead of "figure from Kesti et al."

**Reply:** Changed to *Figure and caption from Kesti et al. (2022).*

---

## Author Response (AR2)

**Replies to reviewer #2 comments:**

**Comment:** The back-trajectory analysis was done for trajectories within the boundary layer only (lines 114ff). This automatically excludes long-range transport of atmospheric constituents in the free troposphere. You also wrote (lines 155f) "The HALO Doppler lidar wind profiles show that there is significant transport aloft (from the direction of the Gulf) throughout the early morning and during the day, which corresponds to the larger mesoscale wind direction (north-westerly flow) for the region which seems to be persistent all year round (citations…).
Q: was this persistent mesoscale wind circulation confirmed by the wind lidar and/or trajectories during the 1-year campaign?

**Reply:** Yes, this persistent mesoscale wind circulation was confirmed by the wind lidar and trajectories during our 1-year campaign.

**Comment:** Why 5 day back trajectories were calculated when the mean wind is 2m/s and the refineries and cities are located within 70-80km (~11 h)? 12h or 24h trajectories seem sufficient. However, in case that the reason is capturing the elevated $SO_2$ concentrations in Iran and Iraq as seen by Tropomi then figure 1 should show a larger area. Why not combining fig. 1 and 11?

**Reply:** No, the trajectory analysis is not restricted to within the boundary layer. Only the back-trajectory starting points were restricted to start within the boundary layer as it says in the text. A back trajectory can well escape the boundary layer, which also expected. This is why the back-trajectory analysis cannot be restricted to e.g. 12 as suggested by the reviewer since transport aloft is more efficient. The fact that we start back-trajectories within the boundary layer, because we have this information, makes the back-trajectory analysis more sound. The reviewer is correct that 5 days is too long to track $SO_2$ in the region and we have changed the wording in the revised manuscript as we just used the first 48 hours of the trajectories in the analysis. 48 hours was chosen as the atmospheric lifetime of $SO_2$ is only a few days in the boundary layer. This was clarified in the text as "A set of  48-hour backward trajectories were computed…" There is no need to say that we calculated 5-day back trajectories when we only used the first 48 hours of them.
We would like to keep Fig. 1 and 11 separate as Fig. 11 with three panels would be quite busy and zooming out Fig. 1 to the same scale and in Fig. 11 would make it impossible to see where the measurement location is on the peninsula.

Minor comment:
**Comment:** Fig.2: please state either in the text or in the figure caption which device was used for the wind roses (Vaisala weather station or Halo wind lidar)

**Reply**: The clarifying text was added to the figure caption: The figure shows data from the automatic weather station.

**Comment:** Fig. 3: the same as for fig. 2

**Reply**: The clarifying text was added to the figure caption: The figure shows data from the automatic weather station.

**Comment:** Fig. 11: the caption of the color scale should have a unit and correspond to the figure caption.

**Reply**: The color scale is unitless. The color scale label was on top of the color bar. The label "RD" is now to the right of the color bar, which is the relative difference (RD) between the different trajectories. The equation for the relative difference is explained in the figure caption.